# AP-4 vesicles contribute to spatial control of autophagy via RUSC-dependent peripheral delivery of ATG9A

Alexandra K. Davies [1], Daniel N. Itzhak [2], James R. Edgar [1], Tara L. Archuleta[3,4], Jennifer Hirst[1], Lauren P. Jackson [3,4], Margaret S. Robinson[1] & Georg H.H. Borner [2]

Adaptor protein 4 (AP-4) is an ancient membrane trafficking complex, whose function has largely remained elusive. In humans, AP-4 deficiency causes a severe neurological disorder of unknown aetiology. We apply unbiased proteomic methods, including 'Dynamic Organellar Maps', to find proteins whose subcellular localisation depends on AP-4. We identify three transmembrane cargo proteins, ATG9A, SERINC1 and SERINC3, and two AP-4 accessory proteins, RUSC1 and RUSC2. We demonstrate that AP-4 deficiency causes missorting of ATG9A in diverse cell types, including patient-derived cells, as well as dysregulation of autophagy. RUSC2 facilitates the transport of AP-4-derived, ATG9A-positive vesicles from the *trans*-Golgi network to the cell periphery. These vesicles cluster in close association with autophagosomes, suggesting they are the "ATG9A reservoir" required for autophagosome biogenesis. Our study uncovers ATG9A trafficking as a ubiquitous function of the AP-4 pathway. Furthermore, it provides a potential molecular pathomechanism of AP-4 deficiency, through dysregulated spatial control of autophagy.

[1] Cambridge Institute for Medical Research, University of Cambridge, Cambridge CB2 0XY, UK. [2] Department of Proteomics and Signal Transduction, Max Planck Institute of Biochemistry, Martinsried 82152, Germany. [3] Department of Biological Sciences, Vanderbilt University, Nashville, TN 37235, USA. [4] Center for Structural Biology, Vanderbilt University, Nashville, TN 37235, USA. These authors contributed equally: Daniel N. Itzhak, James R. Edgar. Correspondence and requests for materials should be addressed to M.S.R. (email: msr12@cam.ac.uk) or to G.H.H.B. (email: borner@biochem.mpg.de)

Eukaryotic cells use a highly regulated system of vesicular and tubular transport intermediates to exchange molecules between organelles. Adaptor protein complex 4 (AP-4) is one of five related heterotetrameric AP complexes, which selectively incorporate transmembrane cargo proteins into nascent vesicles, and recruit machinery for vesicle budding and transport[1]. AP-4 consists of four subunits (β4, ε, μ4 and σ4) forming an obligate complex[2–4] (Fig. 1a). Loss-of-function mutations in any of the genes (AP4B1, AP4E1, AP4M1 and AP4S1) cause a severe recessive neurological disorder with early-onset progressive spastic paraplegia and intellectual disability[5–7]. AP-4-deficient patients have brain abnormalities including thinning of the corpus callosum[6,8–10], indicating an important role for AP-4 in neuronal development and homoeostasis. Axons of Purkinje and hippocampal neurons from Ap4b1 knockout mice contain aberrantly accumulating autophagosomes that are immunopositive for AMPA receptors[11]. However, the link between AP-4 deficiency and dysregulation of autophagy remains unclear.

While the clathrin adaptors AP-1 and AP-2 are well characterised, the function of AP-4, which does not associate with clathrin, has remained elusive. At steady state AP-4 localises to the trans-Golgi network (TGN)[2,3] and so is presumed to mediate cargo sorting at the TGN. The destination of the AP-4 trafficking pathway remains controversial, with conflicting reports suggesting transport to early endosomes[12,13], to late endosomes/lysosomes[14] and, in polarised epithelial cells, to the basolateral membrane[15]. Likewise, AP-4 has been suggested to influence the sorting of various cargoes, including amyloid precursor protein[12,13,16], low-density lipoprotein receptor[11,15], AMPA receptors[11], and δ2 glutamate receptors[11,17]. However, several of these studies relied on exogenously expressed proteins, while the potential endogenous cargoes (e.g., δ2 glutamate receptor) have cell-type-limited expression, unlike AP-4, which is ubiquitously expressed[2,3]. There is currently no consensus as to which proteins are genuine cargoes of the AP-4 pathway, and hence no consensus as to its function. Similarly, AP-4 vesicle machinery is largely uncharacterised. The only identified AP-4 accessory protein is a cytosolic protein of unknown function called TEPSIN[18].

Due to the very low abundance of AP-4 (ca. 40-fold lower than AP-1 or AP-2 in HeLa cells[19]), its functional characterisation has proved challenging. Nonetheless, given its ubiquitous expression in human tissues, AP-4 is likely to play a ubiquitous and important role in protein sorting, whose identification will be paramount to understanding AP-4 deficiency. Here, we combine orthogonal, unbiased and sensitive proteomic approaches to define the composition of AP-4 vesicles. From the intersection of our analyses we identify physiological cargo proteins of the AP-4 pathway, and AP-4 accessory proteins. We demonstrate that AP-4 is required for the correct sorting of three transmembrane cargo proteins, including ATG9A, a protein with a key role in autophagosome biogenesis. Thus, our data suggest a potential mechanistic explanation for the pathology caused by AP-4 deficiency, through dysregulation of autophagy caused by mistrafficking of ATG9A.

## Results

**ATG9A, SERINC1 and SERINC3 are AP-4 cargo proteins**. In AP-4-deficient cells, transmembrane cargo proteins that are usually transported in AP-4 vesicles, and cytosolic accessory proteins that are normally recruited by AP-4, are likely to be mislocalised. As an unbiased screen for such proteins, we used the 'Dynamic Organellar Maps' approach recently developed in our laboratory[19,20]. This mass spectrometry (MS)-based method provides protein subcellular localisation information at the proteome level (Fig. 1b). A comparison of maps made from cells with genetic differences allows the detection of proteins with altered subcellular localisation.

We prepared maps from wild-type, AP4B1 knockout and AP4E1 knockout HeLa cells (Fig. 1c and Supplementary Fig. 1a, b), in biological duplicate (Fig. 1d and Supplementary Data 1). For every protein, we calculated the magnitude of localisation shifts between the wild-type and each knockout, and the reproducibility of the shift direction (Fig. 1e). Three proteins underwent significant and reproducible shifts in both knockout cell lines: SERINC1 and SERINC3 (Serine incorporator 1 and 3), multi-pass membrane proteins of unknown function, and ATG9A (Autophagy-related protein 9A; Fig. 1f). ATG9A is the only transmembrane core autophagy protein and is thought to play a key (though poorly defined) role in the initiation of autophagosome formation[21]. The altered subcellular distribution of these proteins in AP-4-deficient cells identified them as candidate cargo proteins for the AP-4 pathway.

To begin to interpret the nature of the detected shifts, we used subcellular localisation information inferred from the maps. In both wild-type and AP-4 knockout cells, ATG9A and SERINCs mapped to the endosomal cluster (Fig. 1g–i). However, this cluster comprises different types of endosomes, as well as the TGN[19]. Scrutiny of the map visualisations (Fig. 1g–i) and marker protein neighbourhood analysis (Supplementary Data 2) suggested that in the knockouts both SERINCs shifted intra-endosomally, while ATG9A localisation shifted from endosomes towards the TGN.

**RUSC1 and RUSC2 are cytosolic AP-4 accessory proteins**. Cytosolic proteins that only transiently associate with membranes may be missed by the Dynamic Organellar Maps approach, especially if they have low expression levels. We hence applied another proteomic approach developed in our lab, comparative vesicle profiling[18], to identify proteins lost from a vesicle-enriched fraction in the absence of AP-4 (Fig. 2a). This is particularly suited for identifying vesicle coat proteins. Cargo proteins are sometimes less strongly affected, as they may exist in several vesicle populations[18].

We used four different methods to ablate AP-4 function: (i) knockdown; (ii) knocksideways (whereby a protein is rerouted to mitochondria, acutely depleting its cytosolic pool[22]); (iii) knockout of AP4B1; (iv) knockout of AP4E1. Binary comparisons of AP-4-depleted and control vesicle fractions were performed by MS (Supplementary Fig. 2a and Supplementary Data 3) and principal component analysis was used to combine the information from all datasets (Fig. 2b). Proteins that were reproducibly lost from vesicle fractions from AP-4-depleted cells included the AP-4 subunits and TEPSIN, while SERINC1 and SERINC3 were the most depleted membrane proteins. ATG9A was affected to a lesser extent. Two related cytosolic proteins, RUSC1 and RUSC2 (RUN and SH3 domain-containing protein 1 and 2; Fig. 2c), were also reproducibly lost (Fig. 2b). They were not included in the maps since they are barely detectable in subcellular membrane fractions. However, both proteins are highly enriched in the vesicle fraction, suggesting they are vesicle-associated proteins.

To investigate the RUSC-AP-4 relationship, we analysed total membrane fractions and whole cell lysates from control and AP-4 knockout cells by deep sequencing MS. RUSC2 (which has a lower whole cell copy number than RUSC1[19]) was not consistently detected, but RUSC1 was dramatically lost from the total membrane fraction (>4-fold; Fig. 2d) and the whole cell lysate (>3-fold; Supplementary Fig. 1c) in knockout cells

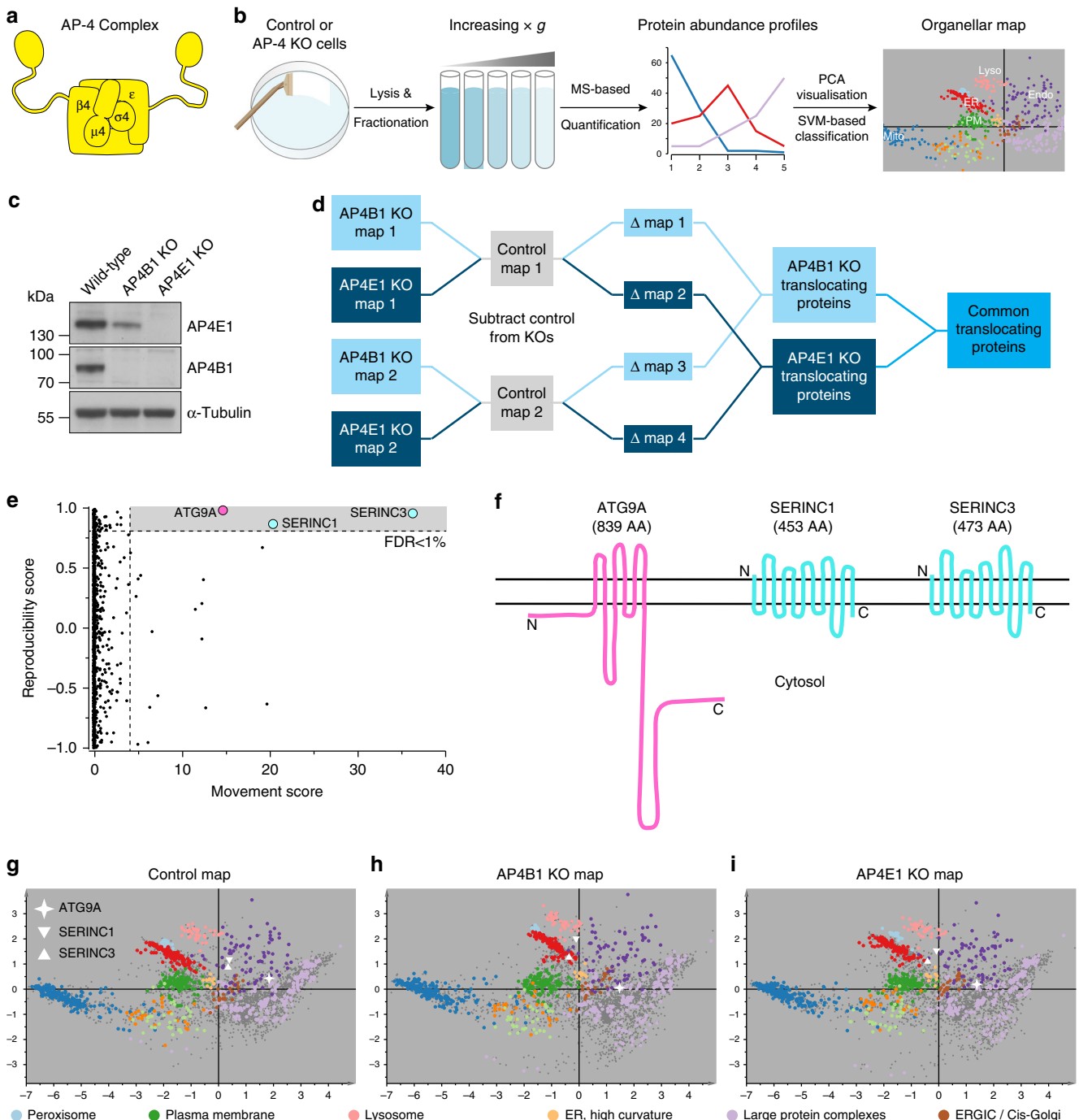

**Fig. 1** Dynamic Organellar Maps detect mislocalisation of ATG9A, SERINC1 and SERINC3 in AP-4 knockout (KO) HeLa cells. **a** Diagram of the AP-4 complex. **b** Workflow for Dynamic Organellar Map generation. Cell lysates are subjected to a series of differential centrifugation steps, to achieve partial separation of organelles. Proteins in each fraction are quantified by mass spectrometry (MS), to obtain abundance distribution profiles. Proteins associated with the same organelle have similar profiles. Clustering can be visualised by principal component analysis (PCA) and compartment assignments are made through support vector machine (SVM)-based classification. **c** Western blot of whole cell lysates from wild-type, *AP4B1* KO and *AP4E1* KO HeLa cells; α-Tubulin, loading control. Representative of two independent experiments. **d** Experimental design for AP-4 Dynamic Organellar Mapping. Maps were made from wild type, *AP4B1* KO and *AP4E1* KO cell lines, each in duplicate. Profiles from each KO map were subtracted from the cognate control profiles, to obtain two AP4E1 Δmaps, and two AP4B1 Δmaps. Proteins that did not shift had similar profiles in wild-type and AP-4 KO maps, and hence Δ profiles close to zero. To identify significantly translocating proteins, the magnitude of shift (M) and the reproducibility of shift direction (R) were scored for each protein and each Δmap. **e** MR plot analysis of AP-4 Dynamic Organellar Mapping. 3926 proteins were profiled across all maps. Three proteins whose subcellular localisation was significantly and reproducibly shifted across the AP-4 KO lines were identified with very high confidence (FDR < 1%). The analysis only covered proteins profiled across all maps; since AP-4 itself was not present in the KO maps, it was not included. See also Supplementary Data 1. **f** Topology of the proteins identified by AP-4 Dynamic Organellar Mapping. **g–i** Visualisation of organellar maps by PCA. Each scatter point represents a protein; proximity indicates similar fractionation profiles. Known organellar marker proteins are shown in colour, and form clusters. Each plot combines the data from two independent map replicates. **g** wild-type; **h** *AP4B1* KO; **i** *AP4E1* KO. The three proteins that undergo significant shifts in AP-4 KOs are annotated

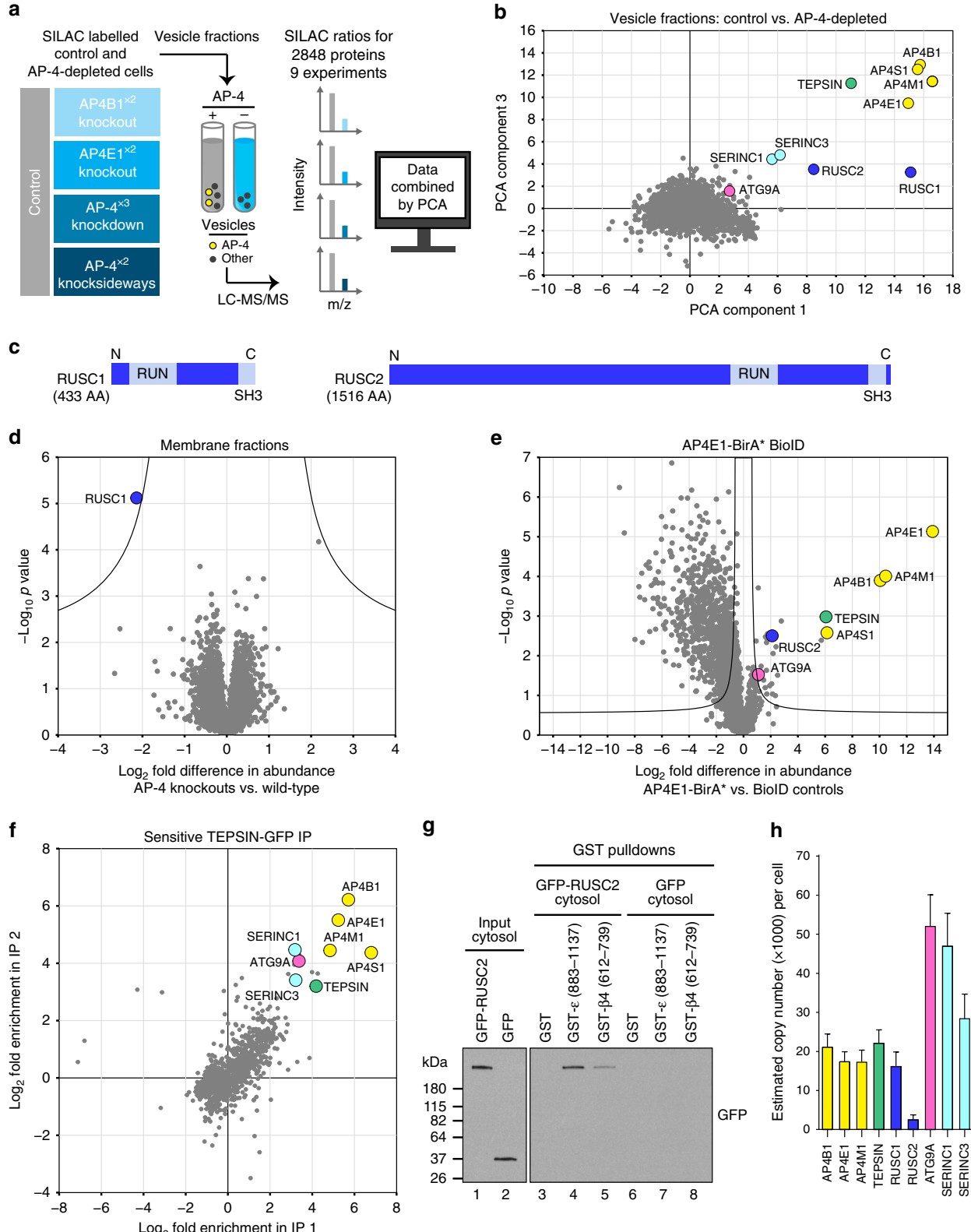

(Supplementary Data 4). This suggests that without AP-4, RUSC proteins are no longer recruited to the membrane, leading to their destabilisation. Mutations in *RUSC2* have been reported in patients with phenotypes reminiscent of AP-4 deficiency[23]. Collectively, these data suggest that RUSC1 and RUSC2 are AP-4 accessory proteins.

**AP-4 interacts with ATG9A, SERINCs and RUSC2.** AP complex-cargo interactions are transient and so have largely proved refractory to standard co-immunoprecipitation approaches. As an alternative we applied BioID, which uses a promiscuous biotin ligase, BirA*, to biotinylate proteins proximal to a protein of interest[24]. We stably expressed AP4E1-BirA* in HeLa

**Fig. 2** Multiple orthogonal proteomic approaches confirm ATG9A, SERINC1 and SERINC3 as AP-4 cargo proteins and identify RUSC1 and RUSC2 as AP-4 accessory proteins. **a** Workflow for proteomic vesicle profiling. Vesicle-enriched fractions were prepared in pairs from metabolically (SILAC heavy or light) labelled control and AP-4-depleted cells and compared by quantitative MS. 2848 proteins were quantified across all experiments. **b** PCA combining the SILAC ratios from nine comparative AP-4-depleted vesicle fraction experiments. Proteins consistently lost from the vesicle fraction of AP-4-depleted cells are in the top-right section. **c** Domain organisation of RUSC1 and RUSC2. There are additional RUSC1 isoforms, but our MS data identified the isoform shown as predominant in our samples. **d** Comparison of protein abundance in total membrane fractions prepared from AP-4 KO (*AP4B1* and *AP4E1*, each in triplicate, n = 6) and wild-type HeLa cells (in triplicate, n = 3), analysed by label-free quantitative MS. >6600 proteins were quantified; RUSC1 was the only protein significantly depleted from the membrane fraction of AP-4 KO cells (RUSC2 and AP-4 subunits were not consistently detected). Data were analysed with a two-tailed *t*-test: volcano lines indicate the significance threshold (FDR = 5%). **e** Comparison of protein abundance in affinity purifications of biotinylated proteins from HeLa cells stably expressing AP4E1-BirA*, and control cell lines (HeLa, HeLa BirA* and HeLa GFP-BirA*), analysed by label-free quantitative MS and volcano analysis as in **d**. The experiment was performed in triplicate and the control dataset was compressed to the three highest LFQ intensities per protein. >3100 proteins were quantified; FDR = 5%. **f** High-sensitivity low-detergent immunoprecipitations from HeLa cells stably expressing the AP-4 associated protein TEPSIN-GFP. The scatter plot shows two replicate SILAC comparisons of TEPSIN-GFP immunoprecipitations versus mock immunoprecipitations (from parental HeLa). TEPSIN-GFP associated proteins have high ratios. **g** Western blots of GST pulldowns using GST-ε (883–1137), GST-β4 (612–739) or GST, from cytosol from HeLa cells stably expressing GFP-RUSC2, or GFP alone as a control. Input cytosol is also shown. Representative of two independent experiments. **h** Estimated copy numbers of AP4B1, AP4E1, AP4M1, TEPSIN, RUSC1, RUSC2, ATG9A, SERINC1 and SERINC3, in HeLa cells, from previously published data[19] (means +/- SD; n = 6)

cells and quantified biotinylated proteins by MS relative to controls (Fig. 2e and Supplementary Data 5). AP4E1-BirA* significantly biotinylated the other three AP-4 subunits and TEPSIN. ATG9A was the only significantly enriched transmembrane protein, consistent with it being an AP-4 cargo protein. SERINC1 and SERINC3 were not identified, likely due to low availability of lysine residues (the target of activated biotin) in their small cytosolic loops[24,25]. In addition, RUSC2 was significantly enriched, supporting its candidacy as an AP-4 accessory protein. Consistent results were obtained with AP4M1-BirA* (Supplementary Fig. 2b–d).

We next performed co-immunoprecipitation of the AP-4 complex via overexpressed TEPSIN-GFP, under sensitive low-detergent conditions (Fig. 2f and Supplementary Data 4). ATG9A, SERINC1 and SERINC3 were co-precipitated, confirming that they are cargo proteins of the AP-4 pathway. As expected, these interactions were not observable with a conventional immunoprecipitation protocol (Supplementary Fig. 2e).

Most AP complex accessory proteins interact with one or both C-terminal "ear" domains of the large subunits, including TEPSIN, which binds to the AP-4 β and ε appendage domains[26,27]. As RUSC2 was identified by BioID, we tested whether GFP-RUSC2 would interact with either of the appendage domains in a GST pull-down experiment. GFP-RUSC2, but not GFP alone, was pulled down with both appendage domains, notably at higher levels with the ε ear (Fig. 2g), confirming that RUSC2 is a *bona fide* AP-4 ear interaction partner.

In sum, using orthogonal proteomic approaches, we have identified three AP-4 cargo proteins, ATG9A, SERINC1 and SERINC3, and two AP-4 accessory proteins, RUSC1 and RUSC2. These are all low abundance proteins, expressed at comparable levels to AP-4 in HeLa cells[19] (Fig. 2h) and primary mouse neurons[20].

**ATG9A accumulates at the TGN of AP-4 deficient cells**. To further characterise the ATG9A missorting phenotype we used immunofluorescence microscopy. In wild-type cells, ATG9A was detected as fine puncta with increased density in the juxtanuclear region (Fig. 3a), consistent with previous data[28]. In contrast, there was a striking accumulation of ATG9A in the TGN region in both AP-4 knockout lines (Fig. 3a and Supplementary Fig. 3). Importantly, the mislocalisation of ATG9A in the *AP4B1* knockout was completely rescued by stable expression of the AP4B1 subunit (Fig. 3a). This was confirmed by quantitative automated imaging (Fig. 3b). To determine whether ATG9A is similarly affected by loss of AP-4 in a cell line more relevant to

the neuronal phenotypes of AP-4 deficiency, we used CRISPR/Cas9-mediated gene editing to deplete AP4B1 or AP4E1 in mixed populations of SH-SY5Y neuroblastoma cells (Fig. 3c). As before, loss of AP-4 caused a striking accumulation of ATG9A at the TGN (Fig. 3d).

To test if ATG9A missorting also occurs in individuals with AP-4 deficiency, we analysed fibroblasts from patients with homozygous mutations in one of the AP-4 genes[5,6,29,30] by immunofluorescence microscopy (Fig. 4a). Mutations in any of the four subunits caused a striking accumulation of ATG9A at the TGN. In addition, fibroblasts from an individual with a heterozygous loss-of-function mutation in *AP4E1* (the phenotypically normal mother of the homozygous *AP4E1* patient) displayed normal ATG9A localisation, so mislocalisation of ATG9A is a cellular phenotype that correlates with disease in AP-4 deficiency.

The microscopy indicated that there was also an increase in the overall ATG9A signal in the patient cells, both within and outside the TGN region. Western blotting confirmed a large increase in the amount of ATG9A in whole cell lysates from all four patient cell lines and an intermediate level in the unaffected *AP4E1* heterozygote (Fig. 4b). These data suggest that in some cases AP-4-deficient cells may compensate for the missorting of ATG9A by increasing its expression, further supporting the importance of AP-4 in the regulation of ATG9A trafficking. However, there was no increase in ATG9A in whole cell lysates from the AP-4 knockout HeLa cell lines (Supplementary Fig. 1c), so the accumulation of ATG9A at the TGN does not simply reflect an increase in expression, but rather a kinetic delay in TGN export in the absence of AP-4.

**ATG9A and SERINCs colocalise in AP-4 dependent vesicles**. Since there are no commercial antibodies that allow detection of endogenous SERINC1 or SERINC3, we used CRISPR to knock in fluorescent Clover tags at the endogenous loci in HeLa cells (Supplementary Fig. 4a). Confocal microscopy with Airyscan enhanced resolution revealed the tagged SERINCs to localise to the perinuclear region and fine puncta throughout the cell (Fig. 5a, c). Strikingly, the peripheral SERINC-positive puncta showed considerable overlap with ATG9A, and AP-4 knockdown resulted in loss of these puncta, suggesting they were AP-4-derived vesicles (Fig. 5a–d and Supplementary Fig. 4b). Furthermore, in the AP-4-depleted cells the SERINCs accumulated in the perinuclear area but, unlike ATG9A, they displayed little colocalisation with TGN46 (Supplementary Fig. 4c, d). Although ATG9A and SERINCs are both cargoes of the AP-4 pathway, they

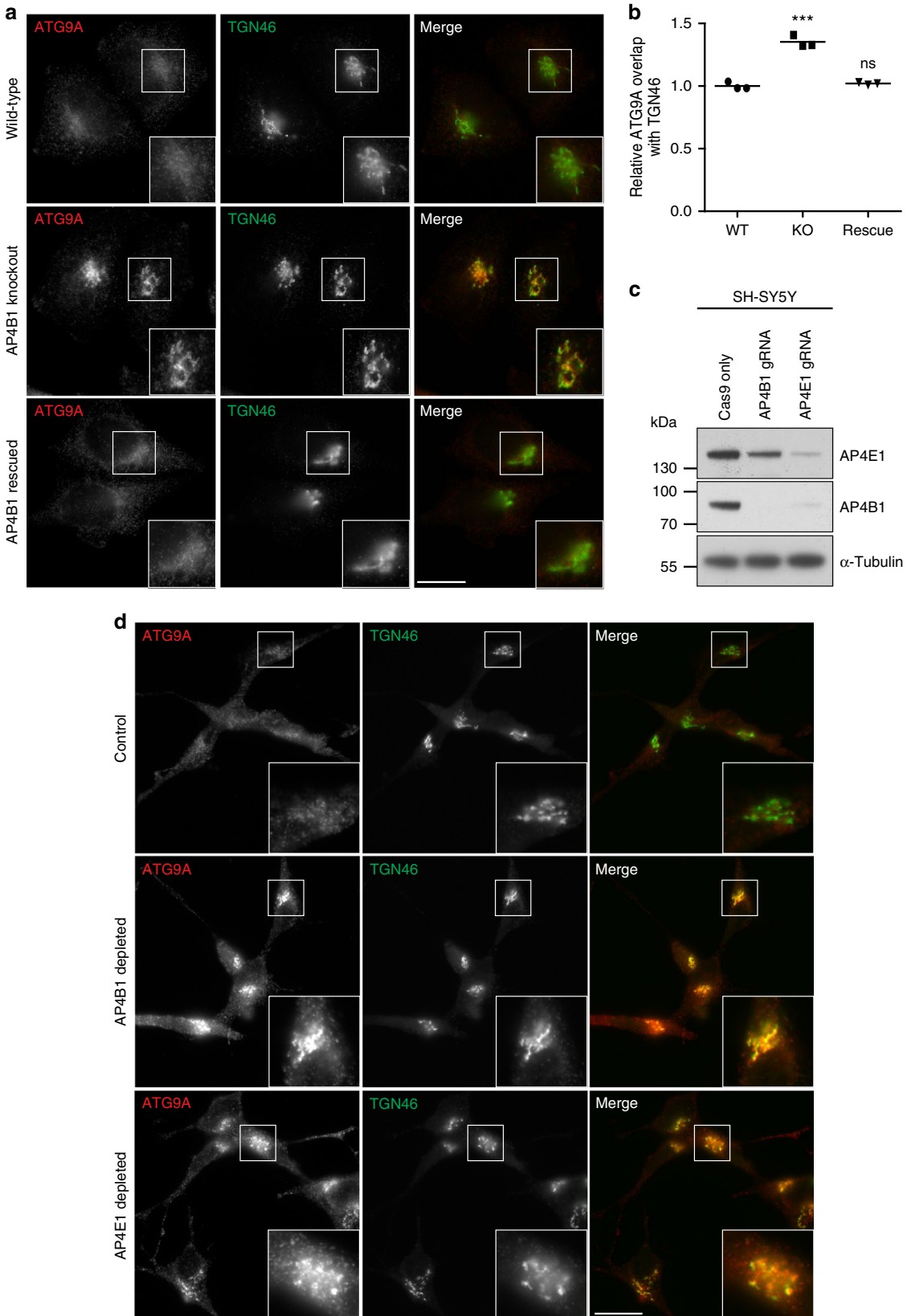

also independently (and perhaps differentially) utilise other trafficking machinery, as evidenced by their partially non-overlapping subcellular distributions in wild-type cells. This may account for different types of perinuclear retention observed in the absence of AP-4.

**RUSC2 drives peripheral accumulation of ATG9A vesicles.** We next created HeLa cell lines that stably overexpress GFP-tagged RUSC2 and found it to localise to fine puncta throughout the cell, with a concentration in clusters at the periphery (Fig. 6a and Supplementary Fig. 5a). Overexpression of RUSC2 resulted in a

**Fig. 3** ATG9A accumulates at the *trans*-Golgi network (TGN) in AP-4 knockout HeLa and SH-SY5Y cells. **a** Widefield imaging of immunofluorescence double labelling of ATG9A and TGN46 in wild-type (WT), *AP4B1* KO, and *AP4B1* KO HeLa cells stably expressing AP4B1 (rescue). Scale bar: 20 μm. **b** Quantification of the ratio of ATG9A labelling intensity between the TGN and the rest of the cell using an automated microscope. Ratios were normalised to the mean wild-type ratio. The experiment was performed in biological triplicate (mean indicated, $n = 3$), and >1400 cells were scored per cell line in each replicate. Data were subjected to one-way ANOVA with Dunnett's Multiple Comparison Test for significant differences from the wild-type: ***$p \leq 0.001$; ns $p > 0.05$. **c** SH-SY5Y (neuroblastoma) cells stably expressing Cas9 were transduced with sgRNAs to *AP4B1* or *AP4E1*. Mixed populations were selected for sgRNA expression and parental Cas9-expressing SH-SY5Y cells were used as a control. Western blot of whole cell lysates; α-Tubulin, loading control. Representative of two independent experiments. **d** Widefield imaging of immunofluorescence double labelling of ATG9A and TGN46 in control, AP4B1 depleted and AP4E1 depleted SH-SY5Y cells. Scale bar: 20 μm

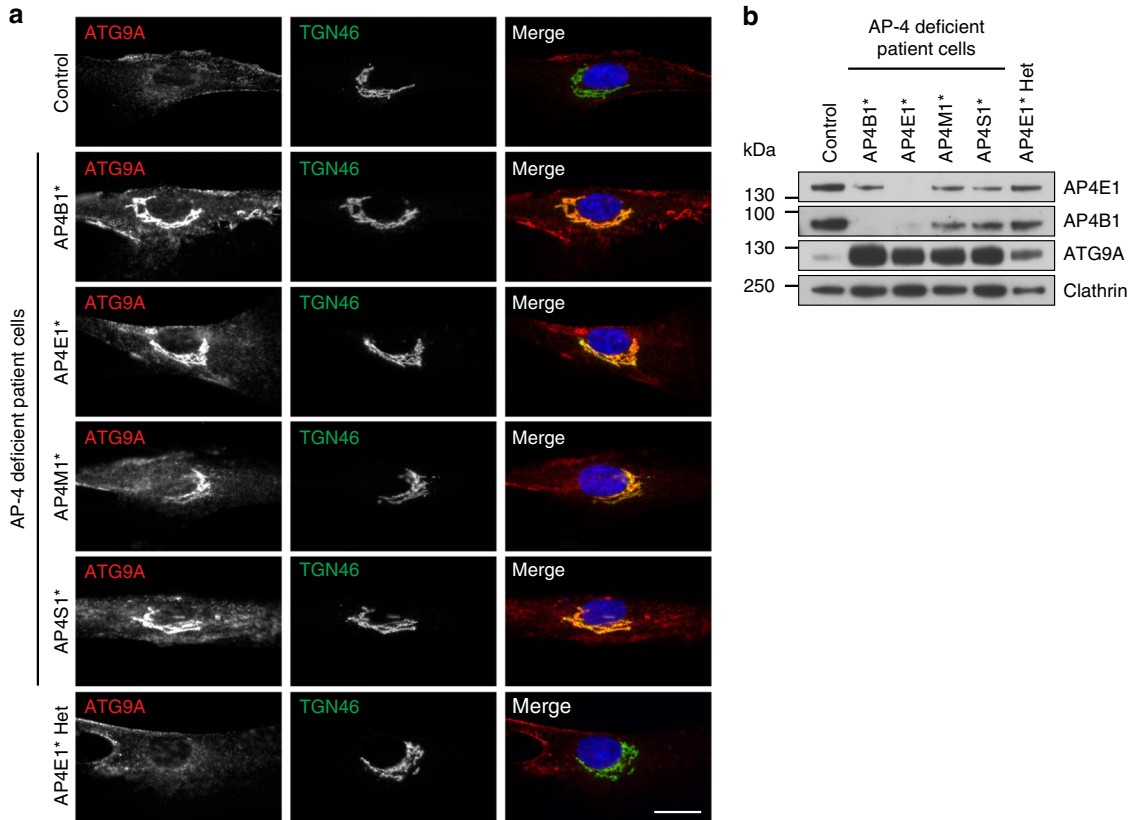

**Fig. 4** ATG9A mislocalisation is a ubiquitous phenotype in cells from AP-4-deficient patients. **a** Widefield imaging of fibroblasts from a healthy control individual, patients with homozygous mutations in one of the four AP-4 genes, and an individual with a heterozygous (Het) mutation in *AP4E1* (phenotypically normal mother of the AP4E1* patient), labelled with anti-ATG9A and anti-TGN46. In the merged image, DAPI labelling of the nucleus is also shown (blue). Scale bar: 20 μm. **b** Western blots of whole cell lysates from the cells shown in **a**; Clathrin heavy chain, loading control. Representative of two independent experiments

dramatic relocation of ATG9A and SERINCs to the cell periphery, where they colocalised with the GFP-tagged RUSC2 (Fig. 6a, b and Supplementary Fig. 5a, b). This effect was specific for AP-4 cargo proteins; other membrane proteins and organellar markers did not respond to overexpression of RUSC2 (Supplementary Fig. 5c).

The localisation of AP-4 itself was not affected by RUSC2 overexpression (Supplementary Fig. 6a), but we hypothesised that the peripheral structures in RUSC2-overexpressing cells might be AP-4-derived vesicles. Consistent with this, GFP-RUSC2 expressed in AP-4 knockout or knockdown cells neither accumulated at the cell periphery, nor colocalised with ATG9A (Fig. 7a and Supplementary Fig. 6b, c). Importantly, both the peripheral distribution of GFP-RUSC2 and its colocalisation with ATG9A were restored in the *AP4B1* knockout by transient expression of the missing AP-4 subunit (Fig. 7a). This demonstrates that the peripheral RUSC2/ATG9A/SERINC-

positive structures are AP-4-dependent compartments. The absence of AP-4 from the accumulating structures suggests that it is released soon after vesicle budding, as is typical of most known vesicle coats[1].

Given the colocalisation between RUSC2 and ATG9A, we tested whether ATG9A and GFP-RUSC2 interact physically. ATG9A co-precipitated with GFP-RUSC2 from wild-type but not *AP4B1* knockout cells (Fig. 7b). This demonstrates that the interaction between RUSC2 and ATG9A (which could be indirect) requires AP-4, suggesting that the three proteins come together transiently during the formation of AP-4 vesicles.

Ultrastructural analysis by correlative light and electron microscopy (CLEM) revealed that the peripheral clusters of GFP-RUSC2-positive puncta corresponded to an accumulation of uncoated vesicular and tubular structures (Fig. 7c). Their peripheral localisation suggested targeting to the plus-ends of microtubules, and RUSCs have been implicated in

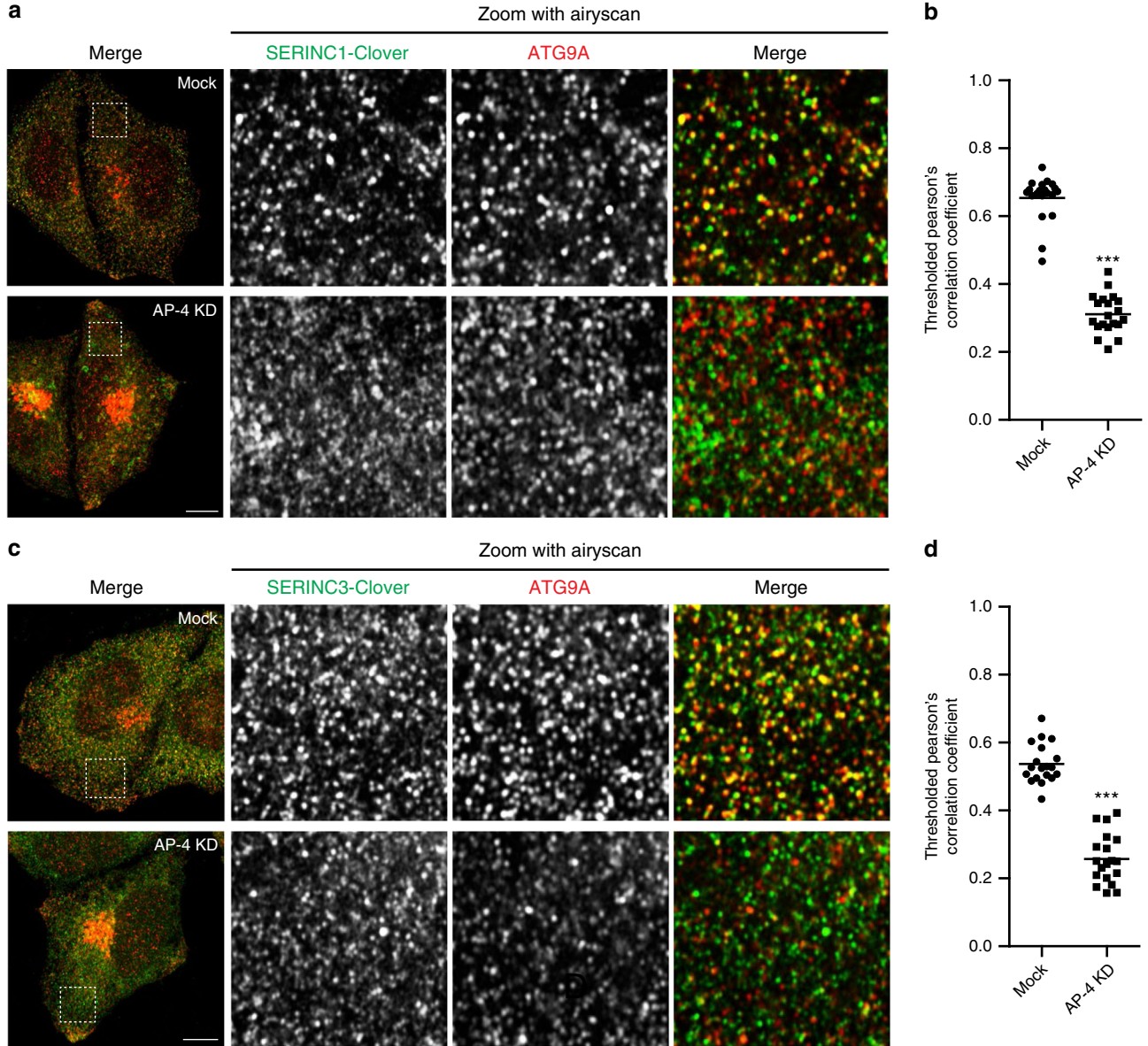

**Fig. 5** ATG9A and SERINC colocalisation in peripheral puncta is dependent on AP-4. CRISPR/Cas9 gene editing was used to introduce a C-terminal Clover (modified GFP) tag to endogenous SERINC1 or SERINC3 in HeLa cells. Cells were transfected with siRNA to knock down AP-4, or were mock transfected (without siRNA) as a control. **a** Confocal microscopy with Airyscanning was used to image SERINC1-Clover (via anti-GFP) and anti-ATG9A. Representative images show a confocal slice of the whole cell, and a peripheral 10 × 10 μm square imaged with Airyscanning in Superresolution mode, used for the quantification of colocalisation between SERINC1-Clover and ATG9A. Scale bar: 10 μm. **b** Quantification of colocalisation between SERINC1-Clover and ATG9A in peripheral regions of mock treated and AP-4 knockdown (KD) cells, using Thresholded Pearson's Correlation Coefficient. 20 cells were quantified per condition. Data show mean ($n = 20$), and results of a two-tailed Mann–Whitney U-test: ***$p \leq 0.001$. **c** Confocal microscopy with Airyscanning was used to image SERINC3-Clover (via anti-GFP) and anti-ATG9A, as in **a**. **d** Quantification of colocalisation between SERINC3-Clover and ATG9A in peripheral regions of mock treated and AP-4 knockdown cells, using Thresholded Pearson's correlation coefficient. 19 cells were quantified per condition. Data show mean ($n = 19$), and results of a two-tailed Mann–Whitney U-test: ***$p \leq 0.001$

microtubule-based transport[31]. Treatment with nocodazole prevented the peripheral localisation of GFP-RUSC2 but, unlike in AP-4-deficient cells, GFP-RUSC2 still co-localised with ATG9A (Fig. 7d). These data suggest that the distribution of AP-4 vesicles requires microtubule-based transport, whereas their formation does not.

**Loss of AP-4 or RUSCs causes dysregulation of autophagy.** *Ap4b1* knockout mice show aberrant accumulation of autophagosomes in neuronal axons, and increased levels of the autophagic marker protein LC3B[11]. We investigated if there were

similar effects on autophagy in our AP-4 knockout HeLa cells. Wild-type and *AP4E1* knockout cells were grown in complete or starvation medium (to induce autophagy), with or without bafilomycin A1 (which blocks autophagosome degradation). The level of LC3B was then assessed by Western blotting (Fig. 8a). In untreated cells, there were increased levels of both unlipidated LC3B-I and lipidated LC3B-II in the *AP4E1* knockout. In addition, the ratio of LC3B-II to LC3B-I was decreased. Under starvation conditions (when LC3B-I is mostly converted into LC3B-II), there was also increased LC3B-II in the *AP4E1* knockout, which increased further in the presence of bafilomycin A1. This

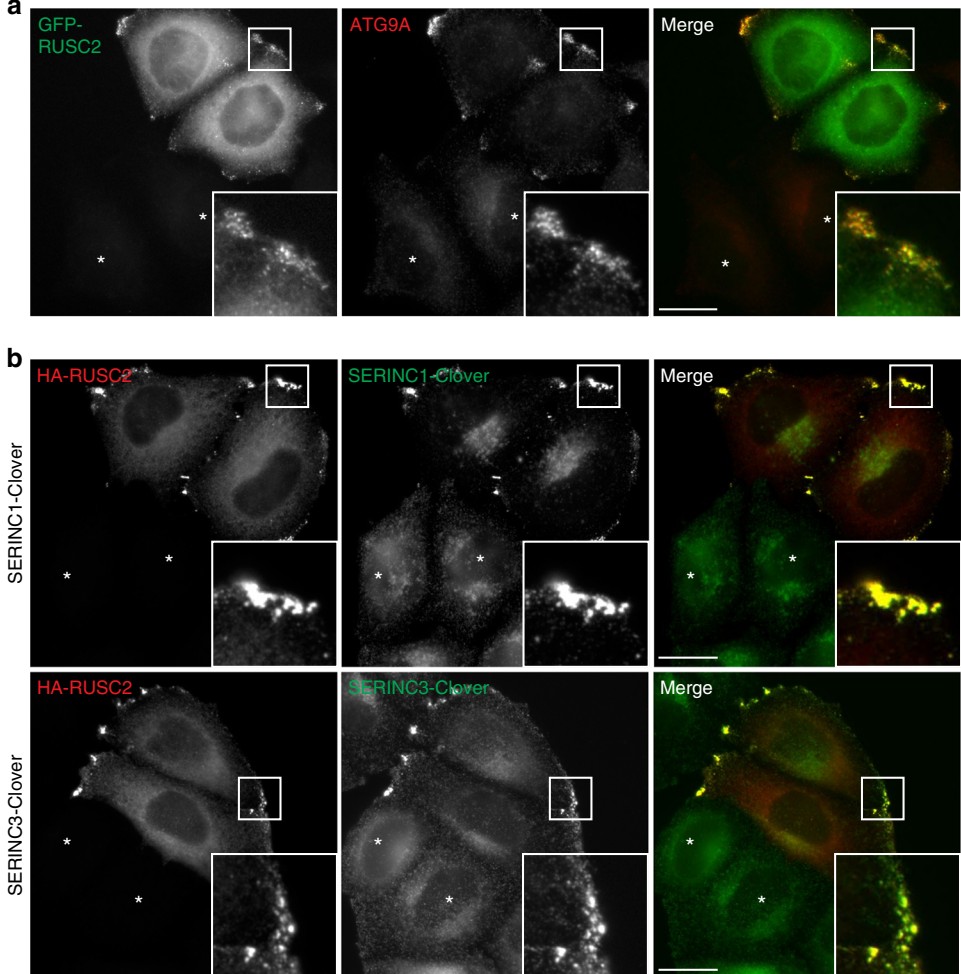

**Fig. 6** ATG9A-positive and SERINC-positive puncta accumulate at the cell periphery in RUSC2-overexpressing cells. **a** Widefield imaging of HeLa cells stably expressing GFP-RUSC2, mixed on coverslips with parental HeLa cells (marked with asterisks), labelled with anti-ATG9A. The insets show accumulation of GFP-RUSC2-positive and ATG9A-positive puncta at the periphery of the cell. Scale bar: 20 μm. **b** Widefield imaging of HeLa cells expressing endogenously tagged SERINC1-Clover or SERINC3-Clover, transiently transfected with HA-RUSC2 and double labelled with anti-GFP and anti-HA. Non-transfected cells are marked with asterisks. The insets show accumulation of HA-RUSC2- and SERINC-positive puncta at the periphery of the cell. Scale bars: 20 μm

suggests that the elevated level of LC3B-II was not due to a block in degradation. The same trends were seen when comparing *AP4B1* knockout cells with the AP4B1 rescued cell line (Fig. 8b). Quantitative MS confirmed the increased level of total LC3B in whole cell lysates of untreated AP-4 knockout cells (median 1.5-fold increase; Supplementary Fig. 1c and Supplementary Data 4).

Elevated LC3B-II may reflect an increase in autophagosome size[32]. To test this, we visualised LC3B-positive structures by immunofluorescence microscopy in wild-type, *AP4B1* knockout, and AP4B1 rescued cells, following two hours starvation (Fig. 8c). Quantification through automated imaging showed an increase in the size of LC3B puncta in the knockout cells (Fig. 8d), but no significant change in the number of puncta (Supplementary Fig. 7a). Larger LC3B puncta were also observed in *AP4E1* knockout cells (Supplementary Fig. 7b). Collectively, these data suggest that lack of AP-4 in HeLa cells causes dysregulation of autophagy.

Given the aberrant autophagy caused by loss of AP-4, we tested whether siRNA-mediated knockdown of either RUSC1, RUSC2, or both together, might also affect autophagy. Transcript levels from both genes were robustly reduced by around 70% (Fig. 8e, f). Knockdown of RUSC1 alone had no effect on the level of LC3B in

HeLa cells grown in complete or starvation medium (Fig. 8g). In contrast, knockdown of RUSC2 resulted in clear alterations to the level of LC3B in both basal and starvation conditions, similar to the effect of loss of AP-4 (Fig. 8a, b, g). Loss of both RUSC proteins had an additive effect, resulting in further elevated LC3B-I and –II levels (Fig. 8g). These data establish RUSCs as important AP-4 accessory proteins and suggest at least partial functional redundancy between RUSC1 and RUSC2.

**ATG9A/RUSC2-positive vesicles associate with autophagosomes**. In contrast to the effect of RUSC2 depletion on autophagy, LC3B levels were unaffected by overexpression of RUSC2, in both basal and starvation conditions (Supplementary Fig. 8a, b). Immunofluorescence microscopy showed similar LC3B-labelling in wild-type and RUSC2-overexpressing HeLa cells grown in full medium (Supplementary Fig. 8c). Interestingly, starvation induced a change in the localisation of RUSC2-positive puncta, away from the periphery of the cell (Fig. 9a and Supplementary Fig. 9a). These puncta were still positive for ATG9A and were often found in very close proximity to LC3B-positive puncta, which appeared larger in the RUSC2-overexpressing cells. Airyscan confocal microscopy revealed the RUSC2-positive and

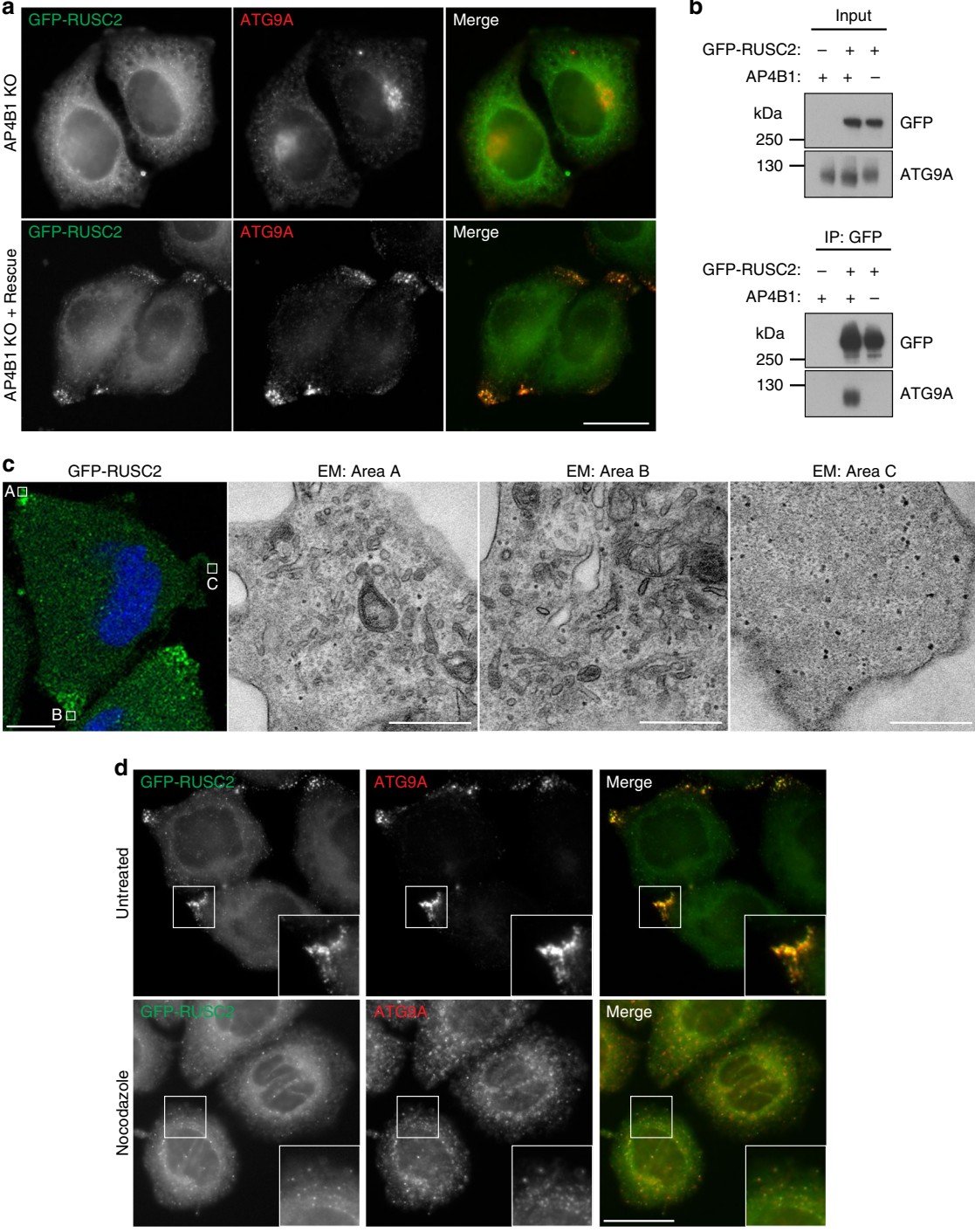

**Fig. 7** RUSC2-driven accumulation of ATG9A-vesicles at the cell periphery depends on AP-4 and microtubules. **a** Widefield imaging of *AP4B1* knockout (KO) HeLa cells stably expressing GFP-RUSC2, labelled with anti-ATG9A, with or without rescue by transient expression of AP4B1. Scale bar: 20 μm. **b** Western blots of immunoprecipitates of GFP-RUSC2 from extracts of wild-type or *AP4B1* KO HeLa cells stably expressing GFP-RUSC2. Parental wild-type HeLa cells were used as a negative control. Representative of three independent experiments. **c** Correlative light and electron microscopy (CLEM) of HeLa cells stably expressing GFP-RUSC2. The peripheral GFP-RUSC2 puncta corresponded to accumulations of small uncoated vesicular and tubular structures (electron micrograph (EM) areas **a** and **b**), which were not found in peripheral regions negative for GFP-RUSC2 (area **c**). Scale bars: fluorescence, 10 μm; EM, 500 nm. **d** Widefield imaging of HeLa cells stably expressing GFP-RUSC2, cultured with or without nocodazole (10 μg mL$^{-1}$, 2 h), labelled with anti-ATG9A. Insets show how disruption of microtubules with nocodazole resulted in loss of the peripheral localisation of GFP-RUSC2 puncta, but their colocalisation with ATG9A remained. Scale bar: 20 μm

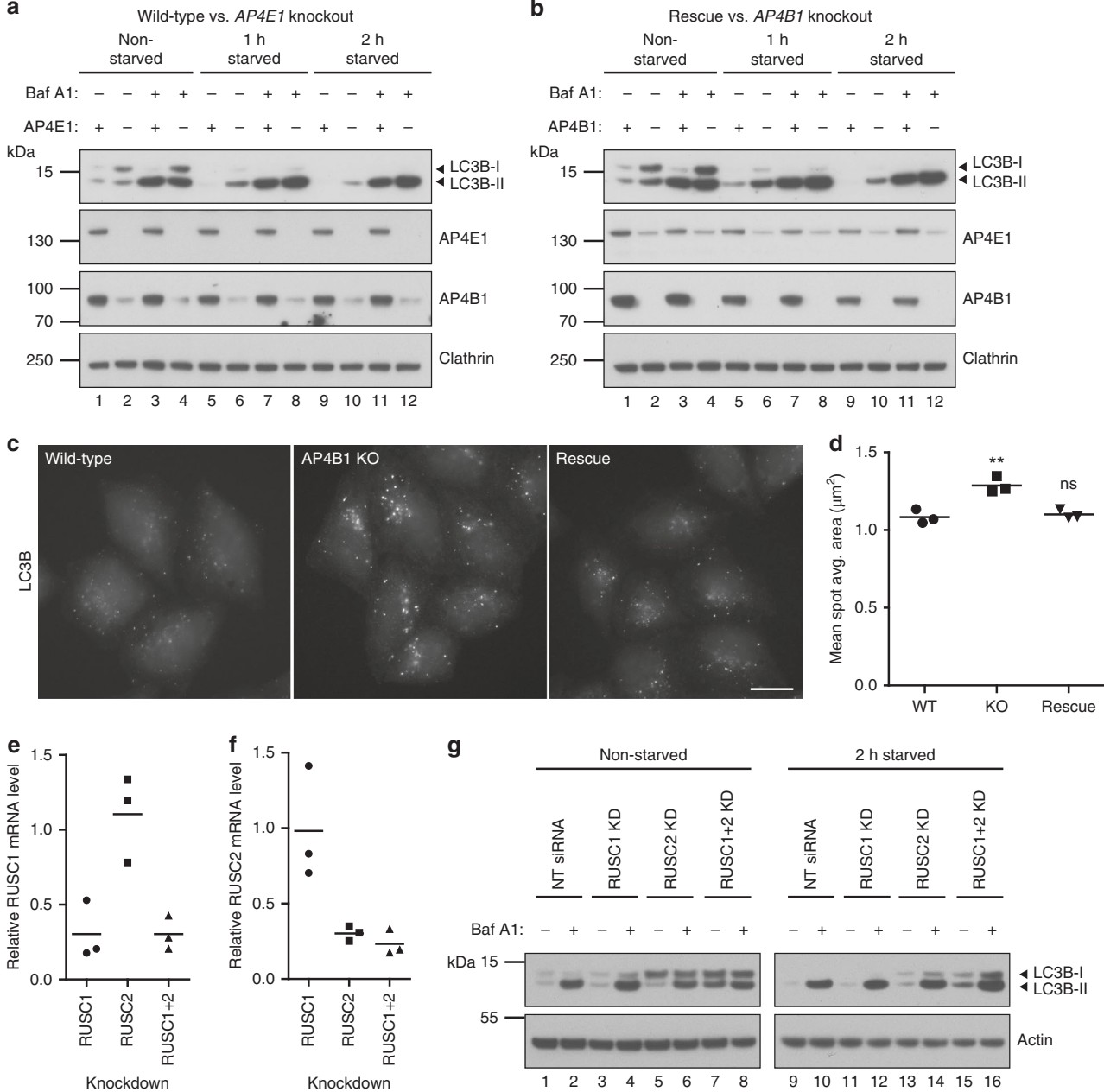

**Fig. 8** Loss of AP-4 or RUSCs in HeLa cells causes dysregulation of autophagy. **a** Western blots of whole cell lysates from wild-type and *AP4E1* KO HeLa cells, cultured in full medium or starved for 1 or 2 h in EBSS, with or without the addition of bafilomycin A1 (Baf A1; 100 nM, 2 h); Clathrin heavy chain, loading control. Representative of two independent experiments. **b** Western blots of whole cell lysates from *AP4B1* KO and *AP4B1* KO HeLa cells rescued with stable expression of AP4B1, as described in **a**. Representative of two independent experiments. **c** Widefield imaging of wild-type, *AP4B1* KO and *AP4B1* KO HeLa cells rescued with stable expression of AP4B1, starved for two hours in EBSS, labelled with anti-LC3B. Scale bar: 20 μm. **d** Quantification of the apparent size (in μm$^2$) of LC3B puncta using an automated microscope. The experiment was performed in biological triplicate (mean indicated, $n = 3$), and over 500 cells were scored per cell line in each replicate. Data were subjected to one-way ANOVA with Dunnett's Multiple Comparison Test for significant differences to the wild-type: **$p \leq 0.01$; ns $p > 0.05$. **e** Quantification by qRT-PCR of *RUSC1* mRNA levels in HeLa cells treated with siRNA to knock down *RUSC1*, *RUSC2*, or both together, relative to cells transfected with a non-targeting (NT) siRNA. The experiment was performed in biological triplicate (mean indicated; $n = 3$). **f** Quantification of *RUSC2* mRNA levels in the same cells, as described in **e**. **g** Western blots of whole cell lysates from HeLa cells treated with siRNA to knock down *RUSC1*, *RUSC2*, both together, or transfected with a non-targeting siRNA, cultured in full medium or starved for two hours in EBSS, with or without the addition of bafilomycin A1 (100 nM, 2 h); Actin, loading control. Representative of two independent experiments

LC3B-positive puncta to be separate structures, which were often closely associated (Fig. 9b). The localisation of GFP-tagged RUSC2 in starved cells was not altered by the presence of bafilomycin A1, but LC3B-positive puncta accumulated under these conditions (Supplementary Fig. 9b). To view these structures at ultrastructural resolution, we performed CLEM of the HeLa

RUSC2-GFP cells grown in starvation conditions in the presence of bafilomycin A1 (Fig. 9c and Supplementary Fig. 9c). The RUSC2-GFP signal observed by light microscopy corresponded to clusters of small, uncoated vesicular structures, as we had previously observed at the periphery of RUSC2-overexpressing cells grown in full medium (Fig. 7c). Importantly, we frequently

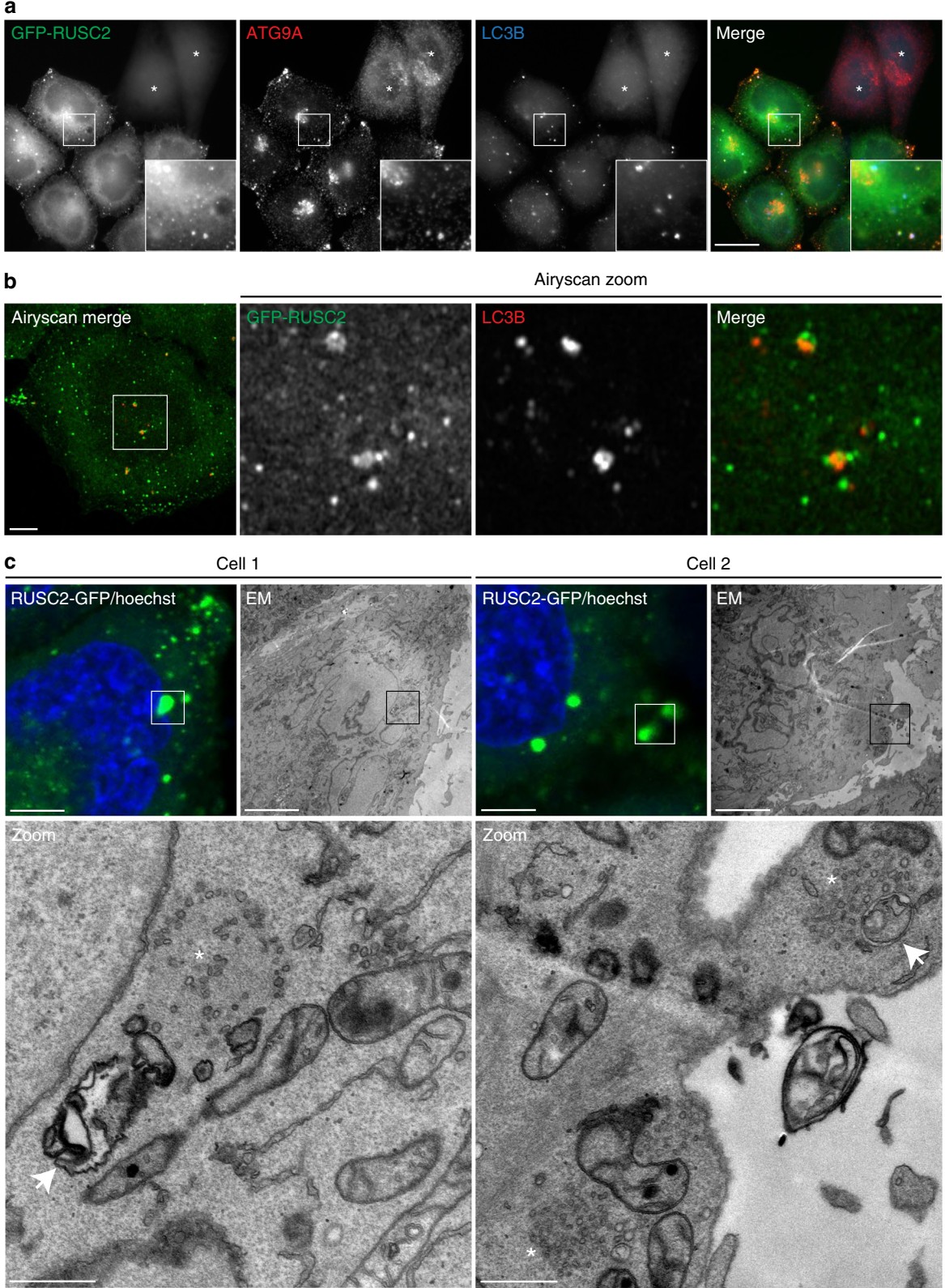

**Fig. 9** RUSC2-positive and ATG9A-positive vesicles closely associate with autophagosomes in starved cells. **a** Widefield imaging of HeLa cells stably expressing GFP-RUSC2, mixed on coverslips with parental HeLa cells (marked with asterisks), starved for two hours in EBSS, labelled with anti-ATG9A and anti-LC3B. The insets show GFP-RUSC2- and ATG9A-positive puncta in very close proximity to LC3B puncta. Scale bar: 20 μm. **b** Confocal imaging with Airyscanning of HeLa cells stably expressing GFP-RUSC2, starved for 2 h in EBSS, labelled with anti-LC3B. The zoomed area is roughly 10 μm². Scale bar: 5 μm. **c** Correlative light and electron microscopy (CLEM) of HeLa cells stably expressing RUSC2-GFP, starved for two hours in EBSS with 100 nM bafilomycin A1. The RUSC2-GFP puncta correspond to clusters of small uncoated vesicular and tubular structures (marked with asterisks), often in close proximity to autophagosomes (marked with arrows). Scale bars: fluorescence and EM, 5 μm; Zoom, 500 nm

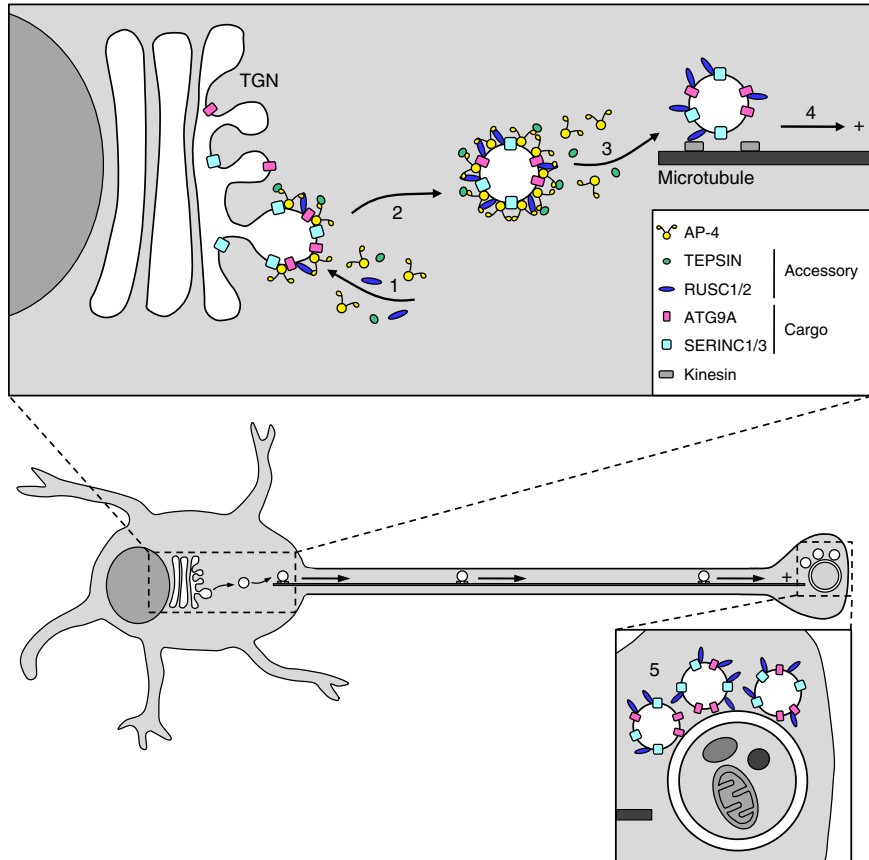

**Fig. 10** Proposed model of AP-4-dependent trafficking. (1) AP-4 and its accessory proteins, TEPSIN, RUSC1 and RUSC2, are recruited to the TGN membrane where they concentrate their transmembrane cargo proteins, ATG9A, SERINC1 and SERINC3, into a vesicle bud; (2) A vesicle carrying ATG9A and SERINCs, coated by AP-4 and its accessory proteins, buds from the TGN membrane; (3) AP-4 and TEPSIN fall off the vesicle membrane and are available for further rounds of vesicle budding at the TGN, while RUSCs remain associated with the vesicle; (4) The vesicle associates with microtubule transport machinery, via the RUSCs, for plus-end-directed transport to the cell periphery; (5) The peripheral ATG9A-containing vesicles act in the nucleation of autophagosomes. In neurons, microtubules are polarised with their plus-ends at the distal axon, which is an important site of autophagosome biogenesis. Thus, in AP-4-deficient neurons ATG9A may not be delivered efficiently to the distal axon, thereby disrupting the spatial control of autophagy

observed double membrane-bound autophagosomes juxtaposed to the RUSC2-positive vesicle clusters, suggesting that the AP-4-derived RUSC2- and ATG9A-positive vesicles are the previously described ATG9A "reservoir" compartment that drives autophagosome biogenesis[21].

## Discussion

The nature of AP-4 vesicles and their role in membrane trafficking has remained elusive for the two decades since their discovery. Here we have used orthogonal global proteomic tools to delineate the function of the AP-4 pathway. We identified three transmembrane cargo proteins, ATG9A, SERINC1 and SERINC3, and three AP-4 accessory proteins, TEPSIN (which had already been identified), RUSC1 and RUSC2. Our approach was hypothesis-free and analysed the subcellular distribution of endogenous proteins. The latter is critical for assessing the role of trafficking pathways, especially for AP-4, which is of comparatively low abundance. When we overexpressed SERINC3, it was no longer trafficked in an AP-4-dependent manner (Supplementary Fig. 4e), suggesting that investigations based on overexpressed candidate cargo proteins are likely to lead to spurious results. The AP-4-associated proteins identified in this study all have low expression levels similar to those of AP-4 itself[19], highlighting the sensitivity of our approach. They are also, like AP-4, expressed ubiquitously. ATG9A localisation depends on

AP-4 not only in HeLa cells, but also in neuroblastoma-derived SH-SY5Y cells and in fibroblasts from AP-4-deficient patients (Fig. 3 and Fig. 4), suggesting that trafficking of ATG9A from the TGN is a ubiquitous function of AP-4. While this manuscript was in preparation, the Bonifacino lab independently identified ATG9A as an AP-4 cargo protein[33], showing accumulation of ATG9A at the TGN in *AP4E1* knockout HeLa and HAP1 cells, and mouse embryonic fibroblasts. In further agreement with our results, they also observed increased levels of ATG9A in AP-4-deficient cells, increased levels of LC3B and enlarged autophagosomal structures. However, the increased sensitivity of our proteomic approaches allowed us to reveal additional interactions between AP-4 and the SERINCs and RUSCs, which were missed by conventional affinity purification approaches.

Our data strongly support a model whereby AP-4 packages ATG9A, SERINC1 and SERINC3 into vesicles at the TGN, which associate via the RUSCs with machinery for microtubule plus-end-directed transport to the cell periphery (Fig. 10). Additionally, the proximity of these vesicles to autophagosomes in starved cells (Fig. 9) suggests a role for AP-4-derived vesicles in autophagosome biogenesis, and hence in the spatial control of autophagy. Neuronal deficiency of Atg9a in mice leads to progressive axonal degeneration, ataxia and convulsions[34]. Thus, provided that AP-4 has equivalent functions in neurons and HeLa cells, which our neuroblastoma data support (Fig. 3d), a hypothesis for

neuronal AP-4 pathology emerges. Neurons require efficient long-range transport, especially towards the distal axon, rendering them susceptible to disturbances in membrane trafficking[35]. Furthermore, microtubules are unipolar in axons, with distally localised plus-ends[36], and the distal axon is an important site of autophagosome biogenesis[37,38]. In *Caenorhabditis elegans* neurons, Atg9-containing vesicles are transported towards the distal axonal microtubule plus-ends, and this is critical for axonal autophagosome biogenesis[39]. Our model suggests that in mammalian neurons, AP-4-derived vesicles carrying ATG9A perform a functionally equivalent role (Fig. 10). In neurons lacking AP-4, ATG9A will not be packaged correctly at the TGN, and will hence not efficiently reach the distal axon. This may interfere with autophagosome formation at the axon terminal and/or with other functions of ATG9A, disrupting neuronal homoeostasis. This model is strongly supported by further work from the Bonifacino lab[40], published during the revision of this manuscript, and by a preprint from the Kittler lab[41], both of which characterise an *Ap4e1* knockout mouse. The mice exhibit a number of neurological motor deficits and brain abnormalities reminiscent of those observed in AP-4-deficient patients. Both groups independently reported an accumulation of Atg9a at the TGN of primary neurons from the Ap4e1-deficient mice, distal axonal swellings, and reduced retrograde movement of autophagosomes, suggesting a defect in autophagosome maturation[40,41].

The identification of ATG9A as an AP-4 cargo, by our lab and others[33,40,41], provides a potential mechanistic basis for the aberrant accumulation of autophagosomes previously observed in neuronal axons of *Ap4b1* knockout mice[11]. There is a similar dysregulation of autophagy in AP-4 knockout HeLa cells, which contain enlarged autophagosomes and increased levels of LC3B (Fig. 8). Basal elevation of LC3B-I and LC3B-II has been observed in *ATG9A* knockout HeLa cell lines[42–44], so our data are consistent with the notion of ATG9A mistrafficking leading to impaired ATG9A function. The role of ATG9A in autophagy is poorly defined, but it is thought to contribute to autophagosome nucleation, without becoming incorporated into the autophagosome membrane itself[21,45]. In yeast the phagophore assembly site originates from Atg9-positive clusters of vesicles and tubules (the "Atg9 reservoir")[46]. Likewise, in mammalian cells Atg9 localises to a tubulovesicular compartment, distinct from other organellar markers[21]. The AP-4-derived ATG9A-positive vesicles and tubules we have observed at the periphery of RUSC2-overexpressing cells fit published descriptions of this compartment[21,46–48] (Fig. 7c). Furthermore, under conditions of starvation these vesicles are found in close proximity to autophagosomes (Fig. 9c), supporting a role in autophagosome biogenesis. In AP-4-deficient cells, ATG9A trafficking may be stalled at the TGN and the peripheral "ATG9A reservoir" depleted. Further investigation is necessary to understand how this may lead to the observed effects on autophagosomes in AP-4-deficient cells. ATG9A mistrafficking has been linked previously to an accumulation of enlarged immature autophagosomes in Niemann-Pick type A patient fibroblasts[49]. The increased size of LC3B puncta in our AP-4 knockout HeLa cells may point to a similar maturation defect, and future CLEM analysis will be important to investigate this further. Finally, it is worth noting that ATG9A has functions independent of autophagy[34,50,51], which may also be relevant to potential pathogenic effects caused by its missorting.

Loss-of-function mutations in *RUSC2* cause a neurological disorder with considerable overlap with the AP-4 deficiency phenotype[23]. This is in keeping with our identification of RUSCs as AP-4 accessory proteins and the very similar effects of AP-4 and RUSC2 depletion on autophagy (Fig. 8). The additive effect of RUSC1 and RUSC2 combined knockdown suggests partial

functional redundancy and links them both clearly to the AP-4 trafficking pathway. Given the role of RUSCs in the subcellular distribution of ATG9A, and hence in the regulation of autophagy, it will be important to further investigate their function, and to identify additional machinery involved. Furthermore, it will be interesting to look for similar ATG9A and SERINC sorting defects in cells from RUSC2-deficient patients. The RUSCs are poorly characterised but implicated in vesicular transport[31,52]. RUSC1 has been proposed to act as a vesicle-transport adaptor by linking syntaxin-1 to kinesin-1 motors[31]. Our data suggest that the RUSCs may similarly link AP-4 vesicles to microtubule transport machinery (Fig. 7).

The presence of SERINC1 and SERINC3 in the ATG9A tubulovesicular compartment also warrants further investigation. Whereas previous studies have relied on overexpressed tagged SERINC proteins for localisation[53,54,56], here we show their endogenous subcellular localisation. Little is known about the function of SERINCs, despite their recent identification as HIV restriction factors[53–55] and their high degree of conservation in all branches of eukaryotes. They were originally proposed to mediate the incorporation of serine into membrane lipids[56], but recent functional studies have shown no effect on membrane composition[57,58]. Our discovery that ATG9A and SERINCs are trafficked together suggests a functional relationship.

In conclusion, this study has greatly expanded our understanding of the AP-4 pathway and provides strong evidence that AP-4-derived vesicles play an important role in the spatial control of autophagy. The identification of AP-4 cargo and accessory proteins provides tools for further investigation of AP-4 function, generates hypotheses about the pathomechanism of AP-4 deficiency, and marks a significant step towards the development of possible treatments.

## Methods

**Antibodies.** The following antibodies were used in this study: rabbit anti-actin 1:3000 for WB (A2066, Sigma-Aldrich), mouse anti-alpha tubulin 1:10,000 for WB (DM1A, T9026, Sigma-Aldrich), rabbit anti-AP4B1 1:400 for WB, rabbit anti-AP4E1 1:1000 for WB (both in-house[3]), mouse anti-AP4E1 1:100 for IF (612019; BD Transduction Labs), rabbit anti-ATG9A 1:1000 for WB and 1:100 for IF (ab108338, Abcam), mouse anti-CIMPR 1:200 for IF (ab2733, Abcam), rabbit anti-clathrin heavy chain 1:10,000 for WB (in-house[59]), mouse anti-EEA1 1:500 for IF (610457, BD Transduction Labs), rabbit anti-GFP 1:500 for IF (gift from Matthew Seaman, University of Cambridge), rabbit anti-GFP 1:1000 for WB (ab6556, Abcam), chicken anti-GFP 1:500 for IF (ab13970, Abcam), mouse anti-HA 1:2000 for WB and IF (16B12, Covance), mouse anti-LAMP1 1:100 for IF (H4A3, Santa Cruz Biotechnology), rabbit anti-LC3B 1:2000 for WB (L7543, Sigma-Aldrich), mouse anti-LC3B 1:400 for IF (M152-3, MBL International), rabbit anti-TEPSIN 1:1000 for WB and 1:250 for IF (in-house[18]), and sheep anti-TGN46 1:200 for IF (AHP500, Bio-Rad). Horseradish peroxidase (HRP)-conjugated secondary antibodies were purchased from Sigma-Aldrich (1:10,000). Fluorescently labelled secondary antibodies used in this study were Alexa488-labelled goat anti-chicken IgY (A11039), Alexa488-labelled donkey anti-mouse IgG (A21202), Alexa555-labelled donkey anti-mouse IgG (A31570), Alexa568-labelled goat anti-mouse IgG (A11031), Alexa594-labelled donkey anti-mouse IgG (A21203), Alexa488-labelled donkey anti-rabbit IgG (A21206), Alexa555-labelled goat anti-rabbit IgG (A21429), Alexa594-labelled donkey anti-rabbit IgG (A21207), Alexa647-labelled donkey anti-rabbit IgG (A31573), Alexa488-labelled donkey anti-sheep IgG (A11015), Alexa594-labelled donkey anti-sheep IgG (A11016), and Alexa680-labelled donkey anti-sheep IgG (A21102), all purchased from Invitrogen and used at 1:500.

**Constructs.** A modified retroviral pLXIN vector (pLXINmod) was a gift from Andrew Peden (University of Sheffield). Myc-BirA* cDNA was amplified from pcDNA3.1_mycBioID (a gift from Kyle Roux; Addgene plasmid #35700[24]) by PCR. *AP4B1* cDNA was amplified from a full-length IMAGE clone (2906087), *AP4E1* cDNA from a full-length IMAGE clone (40146497), and *AP4M1* and *AP4S1* cDNAs from sequence verified EST clones. For generation of the AP-4 knock-sideways construct, a sequence encoding the FKBP domain[22] preceded by a short linker sequence (GALVNGGPEPAKNLYT) was cloned into the natural Sac1 site of the *AP4E1* cDNA in vector pLXINmod, to introduce FKBP into the flexible hinge region of AP4E1 between residues 730 and 731. The control BioID construct, pEGFP-myc-BirA*, was made by cloning myc-BirA* cDNA into a pEGFP-N2 vector (Clontech) using BsrGI and XbaI restriction sites. The AP4E1 BioID

construct was made using Gibson Assembly Master Mix (E2611, New England BioLabs) to introduce myc-BirA* into the flexible hinge region of AP4E1, between residues 730 and 731, inserted into the pLXINmod vector linearised with an HpaI restriction site. *AP4E1* cDNA was amplified as two separate fragments from the AP4E1_FKBP construct described above, including the linker sequence so that it precedes myc-BirA* in the final construct. For AP4B1, AP4M1 and AP4S1 BioID constructs, a C-terminal myc-BirA* tagging construct was used. This was generated using Gibson Assembly to introduce myc-BirA*, preceded by a glycine-serine linker (10 amino acids) and an upstream BglII site, into the HpaI site of pLXIN-mod. The BglII site was used to linearise the myc-BirA* tagging construct and cDNAs for AP4B1/M1/S1 were added by Gibson Assembly. *RUSC2* cDNA was amplified from pCMV-SPORT6_RUSC2 (MHS6278-202800194, Thermo Fisher Scientific) and *GFP* cDNA was amplified from pEGFP-N2. Constructs for stable overexpression of GFP-tagged RUSC2, pQCXIH_GFP-RUSC2 and pQCXIH_-RUSC2-GFP, were generated using Gibson Assembly to introduce *RUSC2* and *GFP* cDNAs into the retroviral vector pQCXIH (Clontech). The AgeI site of pQCXIH was used for GFP-RUSC2 and the NotI site for RUSC2-GFP. pQCXIH_HA-RUSC2 was made by cutting out *GFP* from pQCXIH_GFP-RUSC2 with NotI and AgeI restriction sites and replacing it with a triple HA tag. *SERINC3* cDNA was amplified from pIRESNeo2_SERINC3-HA-mCherry, a custom synthetic construct (Genecust) based on the *SERINC3* clone AAD22448.1, with several silent nucleotide substitutions. The construct for stable overexpression of HA-tagged SERINC3, pLXIN_SERINC3_HA, was made using Gibson Assembly to introduce an HA tag between residues 311 and 312 of SERINC3 (within an extracellular loop), inserted into the pLXINmod vector linearised with HpaI. The sequences of all constructs described here were verified by Sanger DNA sequencing. See Supplementary Methods for a complete list of the PCR primers used.

**Cell culture**. HeLa M cells and HEK 293ET cells were from ECACC and the human neuroblastoma cell line SH-SY5Y[60] was from Sigma-Aldrich. The HeLa cells stably expressing BirA* were a gift from Folma Buss (University of Cambridge) and the HeLa cells stably expressing EGFP were a gift from Matthew Seaman (University of Cambridge). The HeLa cells stably expressing TEPSIN-GFP used for the conventional and sensitive immunoprecipitations had previously been generated in our laboratory[18]. The *AP4B1* knockout HeLa cells (clone x2A3) and wild-type β4 rescued *AP4B1* knockout HeLa cells were previously described[26]. The human fibroblasts from AP-4 deficient patients have been previously described and reduced AP-4 complex formation has been demonstrated for all patient lines. Molecular details of mutations and references for each are as follows: *AP4B1**, GenBank NM_006594: c.487_488insTAT, p.Glu163_Ser739delinsVal[5,18] (a gift from Laurence Colleaux, Institut Imagine, and Annick Raas-Rothschild, Tel Aviv University); *AP4E1**, GenBank NM_007347: c.3313C>T, p.Arg1105*[29] (a gift from Xiao-Fei Kong, Jean-Laurent Casanova, and Stephanie Boisson-Dupuis, The Rockefeller University); *AP4M1**, GenBank NM_004722: c.1137+1G>T, p.Ser342ArgfsTer65[6,18] (a gift from Grazia Mancini, Erasmus Medical Center); *AP4S1**, GenBank NM_007077: c.[289C>T];[138+3_6delAAGT], p.[Arg97*];[?][30] (the second mutation, which is at a splice donor site, was originally inaccurately reported as c.138_140delAATG). The control fibroblasts (from a healthy control donor) were a gift from Craig Blackstone (NIH) and the heterozygous *AP4E1*WT/ *AP4E1** fibroblasts (a gift from Kong et al. as above) were from the unaffected mother of the homozygous *AP4E1** patient.

HeLa M cells were maintained in RPMI 1640 (R0883, Sigma-Aldrich). SH-SY5Y cells and human fibroblasts were maintained in Dulbecco's Modified Eagle's Medium (DMEM) high glucose (D6546, Sigma-Aldrich). All media was supplemented with 10% v/v foetal calf serum, 4 mM L-glutamine, 100 U mL$^{-1}$ penicillin and 100 µg mL$^{-1}$ streptomycin and all cells were cultured at 37 °C under 5% $CO_2$. Stable cell lines were additionally maintained with 500 µg mL$^{-1}$ G418 or 150 µg mL$^{-1}$ hygromycin as appropriate. For metabolic labelling (SILAC method[61]) in most experiments, HeLa cells were cultured in DMEM without Arginine, Glutamine, Lysine or Sodium Pyruvate (A14431-01, Gibco), supplemented with 10% (v/v) dialysed foetal calf serum (A11-107, PAA), 1 mM Sodium Pyruvate (58636, Sigma-Aldrich), 1x GlutaMAX (35050-061, Gibco), and either "Heavy" amino acids (42 mg L$^{-1}$ $^{13}C_6$, $^{15}N_4$-L-Arginine HCl and 73 mg L$^{-1}$ $^{13}C_6$, $^{15}N_2$-L-Lysine HCl; 201604302 and 211604302, Silantes), or the equivalent "Light" amino acids (Arginine HCl [A6969] and Lysine HCl [L8662], Sigma-Aldrich). Cells were cultured for at least seven days in these media before experiments were performed. In the vesicle fraction experiments, metabolic labelling was performed in SILAC RPMI 1640 medium (89984, Thermo Fisher Scientific), supplemented with 10% (v/v) dialysed foetal calf serum (10,000 MW cut-off; Invitrogen), and either "Heavy" amino acids (50 mg L$^{-1}$ $^{13}C_6$, $^{15}N_4$-L-Arginine HCl and 100 mg L$^{-1}$ $^{13}C_6$, $^{15}N_2$-L-Lysine 2HCl; Cambridge Isotope Laboratories), or the equivalent "Light" amino acids.

Transient DNA transfections were carried out using a TransIT-HeLaMONSTER® kit (Mirus Bio LLC), according to the manufacturer's instructions. Stable cell lines were created using retrovirus made in HEK 293ET cells transfected using TransIT-293 Transfection Reagent (Mirus Bio LLC), according to the manufacturer's instructions. pLXIN or pQCXIH plasmids were mixed with the packaging plasmids pMD.GagPol and pMD.VSVG in a ratio of 10:7:3. Viral supernatants were harvested after 48 h, filtered through a 0.45 µm filter, supplemented with 10 µg mL$^{-1}$ hexadimethrine bromide (Polybrene, Sigma-

Aldrich) and applied directly to the target cells at 37 °C. Antibiotic selection for stable expression (500 µg L$^{-1}$ G418 or 150 µg mL$^{-1}$ hygromycin) was initiated 48 h post-transduction. When necessary due to variable levels of transgene expression in mixed populations of stably transduced cells, cell lines were single cell cloned by serial dilution. For generation of the AP-4 knocksideways cell line, the AP4E1-FKBP construct was introduced into HeLa cells stably expressing a mitochondrially targeted bait (Mito-YFP-FRB[22]), using retrovirus as described above. Clonal cell lines were isolated and selected for appropriate expression levels. Correct localisation of the AP4E1-FKBP and rapamycin-induced rerouting of AP-4 to mitochondria were verified by microscopy. HeLa cells stably overexpressing SERINC3-HA-mCherry were generated by plasmid (pIRESNeo2_SERINC3-HA-mCherry) transfection and antibiotic selection in G418. A mixed population of medium-level overexpressing cells was obtained through FACS. Importantly, due to the overexpression, the subcellular sorting of tagged SERINC3 is no longer dependent on AP-4 in these cells (Supplementary Fig. 4e).

Where indicated cells were treated with 10 µg mL$^{-1}$ nocodazole in cell culture medium for 2 h at 37 °C. For starvation during autophagy assays, cells were washed three times with Earle's balanced salt solution (EBSS; Sigma-Aldrich) and incubated in EBSS for the specified time. Where indicated cells were treated with 100 nM Bafilomycin A1 in cell culture medium or EBSS for 2 h. For AP-4 knocksideways, cells were treated with rapamycin at 200 ng mL$^{-1}$ final concentration (from a 1 mg mL$^{-1}$ stock in ethanol) for 60 min.

No cell lines used in this study were found in the database of commonly misidentified cell lines that is maintained by ICLAC and NCBI Biosample. The cell lines were routinely tested for the presence of mycoplasma contamination using DAPI to stain DNA and were regularly treated with mycoplasma removing agent (093050044, MP Biomedicals).

**CRISPR/Cas9-mediated gene editing**. The Zhang online CRISPR design tool (http://crispr.mit.edu/[62]) was used to identify suitable gRNA targets. gRNAs were ordered as pairs of complementary oligos (Sigma-Aldrich) with the sequences 5′-CACCGN20-3′ and 5′-AAACN20C-3′, which were annealed and cloned into the BbsI site of the appropriate Cas9/gRNA delivery vector.

For the *AP4E1* knockout HeLa cell line we inactivated all copies of the *AP4E1* gene using the 'double nickase' CRISPR/Cas9 system[63,64]. Paired gRNAs targeting exon 6 of *AP4E1* (ENST00000261842: CTTGATTAGGAGCAATGAGA and GCACTTTGTGACAGAGATGT) were individually cloned into pX335-U6-Chimeric_BB-CBh-hSpCas9n(D10A) vectors (a gift from Feng Zhang; Addgene plasmid #42335[63]). HeLa M cells were transfected with both pX335 plasmids and pIRESpuro (Clontech) in a ratio of 2:2:1. Forty-eight hours later, untransfected cells were killed off by a 4-day selection in 1 µg mL$^{-1}$ puromycin. Single cell clones were isolated and tested for knockout of AP4E1 by Western blotting and immunofluorescence. Clone x6C3 was negative for AP4E1 expression in both assays and was further validated by sequencing. Genomic DNA was harvested using a High Pure PCR Template Purification Kit (Roche) and PCR was used to amplify the region around the target sites. The PCR products were then blunt-end cloned (Zero Blunt PCR Cloning Kit; Invitrogen) and 17 clones were sent for Sanger sequencing with the M13_F primer (Beckman Coulter Genomics).

For depletion of *AP4B1* and *AP4E1* in SH-SY5Y cells, a lentiviral CRISPR/Cas9 system was used[65]. Lentivirus was produced by transfecting HEK 293ET cells with the lentiviral vector plus the packaging plasmids pCMVΔR8.91 and pMD.G in a ratio of 10:7:3, using TransIT-293 Transfection Reagent. Harvest of viral supernatants, lentiviral transductions and selection for stable expression (150 µg mL$^{-1}$ hygromycin or 3 µg mL$^{-1}$ puromycin) were performed as described above for retroviral transductions. Wild-type Cas9 was stably introduced into SH-SY5Y cells using the lentiviral vector pHRSIN-P$_{SFFV}$-FLAG-Cas9-P$_{PGK}$-Hygro (a gift from Paul Lehner, University of Cambridge). Cas9-expressing SH-SY5Y cells were then transduced with gRNAs targeting exon 2 of *AP4B1* (ENST00000369569.5: GACCCCAATCCAATGGTGCG) or exon 6 of *AP4E1* (ENST00000261842.9: GCACTTTGTGACAGAGATGT), cloned into the lentiviral sgRNA expression vector pKLV-U6gRNA(BbsI)-PGKpuro2ABFP (a gift from Kosuke Yusa; Addgene #50946[66]). Mixed populations of cells were selected for stable expression of Cas9 and gRNA and knockouts were assessed at the protein level by Western blotting.

For CRISPR/Cas9-mediated endogenous tagging of SERINC1 and SERINC3, we used homology-directed repair to introduce a myc-Clover tag at the C-terminus of each protein. Suitable sgRNA targets were selected to enable Cas9 to cut downstream and proximal to the STOP codon of *SERINC1* (ENST00000339697.4: ATACACAACTTTACAAAAGT) and *SERINC3* (ENST00000342374.4: GGTATATGGGTTTTCGGTGA). gRNAs were cloned into pX330-U6-Chimeric_BB-CBh-hSpCas9 vectors (a gift from Feng Zhang; Addgene plasmid #42230[63]), containing cDNA for wild-type Cas9. To generate the homology-directed repair plasmids we made use of a pDonor_myc-Clover plasmid which was a gift from Dick van den Boomen and Paul Lehner (University of Cambridge). This was created using a plasmid gifted to them by Matthew Porteus and Ron Kopito (Stanford University), originally containing a TAP-tag, which was replaced with myc-Clover cloned from pcDNA3.1-Clover-mRuby2 (a gift from Kurt Beam; Addgene #49089[67]). pDonor_myc-Clover contains a 5′ homology region, followed by myc-Clover, an internal ribosome entry site, a puromycin resistance gene and then a 3′ homology region. The existing homology regions were replaced by Gibson Assembly with regions specific for *SERINC1* (ENSG00000111897: 5′ – 806 bp preceding the STOP codon; 3′ – 814 bp starting 161 bp after the STOP codon) or

*SERINC3* (ENSG00000132824: 5′ – 835 bp preceding the STOP codon; 3′ – 817 bp starting 95 bp after the STOP codon). 3′ homology regions were chosen to avoid the gRNA target sites. The primers used to clone the homology regions are provided in Supplementary Methods. HeLa M cells were transfected with the pX330 and pDonor plasmids in a ratio of 1:1. Forty-eight hours later selection for stable expression of the puromycin resistance gene (meaning incorporation of the Clover tag) was initiated. Single cell clones were isolated and tested for knock-in of Clover by Western blotting and immunofluorescence with an anti-GFP antibody. Clones SERINC1-Clover A3 and SERINC3-Clover B6 were positive for Clover expression in both assays and correct integration of the tag was confirmed by Sanger DNA sequencing.

**siRNA-mediated knockdown**. Knockdown of AP-4 was achieved by combined siRNA targeting of *AP4E1* and *AP4M1* using ON-TARGETplus SMARTpools (*AP4E1*, L-021474-00; *AP4M1*, L-011918-01; Dharmacon), using a double-hit 96 h protocol[18]. For the first hit the final concentration of siRNA was 40 nM (20 nM *AP4M1* + 20 nM *AP4E1*). The second hit was performed 48 h after the first hit with half the final concentration of siRNA. For knockdown of RUSC1 an ON-TARGETplus SMARTpool (L-020607-01, Dharmacon) was used. For knockdown of RUSC2 two independent GeneSolution siRNAs (Hs_RUSC2_4, SI00709128; Hs_RUSC2_7, SI04296264; Qiagen) were used, resulting in similar knockdown efficiencies at the mRNA level and highly similar phenotypes in our LC3B Western blot assay. For clarity, only the data for one oligo (Hs_RUSC2_4) is shown in Fig. 8, but the Western blot for Hs_RUSC2_7 can be seen in the uncropped scan of the blot shown in Supplementary Fig. 11. For the RUSC knockdowns a double-hit 72 h protocol was used with a final siRNA concentration of 20 nM for the first hit and 10 nM for the second hit, performed 36 h later. For the combined RUSC1 and 2 knockdowns the final concentration of siRNA was 20 nM for each gene (10 nM for the second hit). Transfections of siRNA were carried out with Oligofectamine (Thermo Fisher Scientific), according to the manufacturer's instructions and where indicated cells were mock treated with Oligofectamine without siRNA or transfected with ON-TARGETplus Non-targeting siRNA #1 (D-001810-01, Dharmacon). Three independent experiments were performed, unless otherwise noted in the figure legend.

**Quantitative RT-PCR**. For quantitative RT-PCR analysis, total RNA was extracted from around $5 \times 10^5$ cells using an RNeasy Mini Kit (Qiagen) with on-column DNase digestion using an RNase-Free DNase Set (Qiagen), according to the manufacturer's instructions. A total of 500 ng of total RNA was reverse-transcribed using TaqMan® Reverse Transcription Reagents (Thermo Fisher Scientific) in a 20 μL reaction, according to the manufacturer's instructions. Quantitative-PCR was performed using TaqMan® gene expression assays (Thermo Fisher Scientific), which include two unlabelled PCR primers and one FAM™ dye-labelled TaqMan® MGB probe, according to the manufacturer's instructions; TaqMan® Assay IDs were Hs00204904_m1 for *RUSC1* and Hs00922017_m1 for *RUSC2*. The cDNA was diluted 1:5, and 5 μL of this dilution (25 ng RNA equivalent) was used for each PCR reaction using TaqMan® Universal PCR Master Mix (Thermo Fisher Scientific) and using a 7900HT Fast Real-Time PCR System (Thermo Fisher Scientific), with Standard mode thermal cycling conditions. Every reaction was carried out in technical triplicate and each knockdown was performed in biological triplicate. For each sample, the levels of *RUSC1* and *RUSC2* mRNAs were normalised using *GAPDH* as a loading control (TaqMan® Assay ID Hs99999905_m1; Thermo Fisher Scientific). The data were analysed in Microsoft Excel using the ΔΔCt method, relative to a non-targeting siRNA control. Results are expressed as means ± SEM for each biological triplicate.

**Fluorescence microscopy**. Cells were grown onto 13 mm glass coverslips and fixed in 3% formaldehyde in PBS (137 mM NaCl, 2.7 mM KCl, 10 mM $Na_2HPO_4$ and 1.76 mM $KH_2PO_4$, pH 7.4) or, for AP4E1 labelling, ice-cold methanol. Formaldehyde fixed cells were permeabilised with 0.1% saponin and blocked in 1% BSA/0.01% saponin in PBS. Methanol fixed cells were blocked in 0.5% BSA. Primary antibody (diluted in BSA block) was added for 45 min at room temperature. Coverslips were washed three times in BSA block and then fluorochrome-conjugated secondary antibody was added in block for 30 min at room temperature. Coverslips were then washed three times in PBS, followed by a final wash in $dH_2O$, before being mounted in ProLong Diamond Antifade Reagent with DAPI (Thermo Fisher Scientific). Widefield images were captured on an Axio Imager II microscope (63×/1.4 NA oil immersion objective; AxioCam 506 camera) and confocal and Airyscan enhanced resolution images were captured on an LSM880 confocal microscope with Airyscan (63×/1.4 NA oil immersion objective; ZEISS), both equipped with ZEN software (ZEISS). Airyscan images were taken in SR (super resolution) mode and raw data were processed using Airyscan processing in "auto strength" mode (strength = 6.0) with Zen Black software version 2.3. Quantification of SERINC1/3-Clover and ATG9A colocalisation was performed on Airyscan images. Colocalisation was measured using Pearson's Correlation Coefficient with Costes thresholding method (Volocity software 6.3; Perkin Elmer), in a peripheral 10 μm² area in each cell, selected while viewing the green channel only. A minimum of 19 cells were analysed for each condition. For statistical analysis data were analysed by a two-tailed Mann–Whitney *U*-test.

For automated imaging of ATG9A localisation and LC3B puncta, cells were plated in 96-well microplates (6005182, Perkin Elmer), which were either uncoated or for the LC3B assay coated with poly-D-lysine. Cells were fixed in 3% formaldehyde in PBS, permeabilised with 0.1% saponin and labelled for immunofluorescence as described above. After washing off secondary antibody, cells were stained with HCS CellMask™ Blue stain (Thermo Fisher Scientific) diluted 1:5000 in PBS for 30 min at room temperature. Cells were then washed three times in PBS before imaging using a CellInsight CX7 High-Content Screening Platform (Olympus 20×/0.4NA objective; Thermo Fisher Scientific) running HCS Studio™ 3.0 software. Autofocus was applied using the whole cell mask channel (channel 1). Experiments were performed in biological triplicate with a technical triplicate (three separate wells per cell line) within each experiment. ATG9A localisation was quantified using the Colocalization Bioapplication V4 (Cellomics, Thermo Fisher Scientific), using anti-TGN46 to segment the TGN (channel 2; ROI_B). ROI_A was defined by the whole cell mask (channel 1) minus ROI_B. The average intensity of anti-ATG9A (channel 3; target 1) was then quantified in the two regions and a ratio calculated between the two. Ratios were normalised to the mean wild-type ratio. At least 1400 cells were scored per cell line in each experiment. For statistical analysis data were log transformed and analysed by one-way ANOVA with Dunnett's Multiple Comparison Test. LC3B puncta were quantified using the Spot Detector Bioapplication V4 (Cellomics, Thermo Fisher Scientific). Spots were identified with smoothing on (uniform; value = 1), with the detection method Box (value = 1) and ThreeSigma thresholding (value = 0.012). Spot total count and average area (in μm²) were measured. At least 500 cells were scored per cell line in each experiment. For statistical analysis data were analysed by one-way ANOVA with Dunnett's Multiple Comparison Test.

Where representative images are shown, the experiment was repeated at least two times. Statistical analyses of imaging data were performed using GraphPad Prism version 5.01 (GraphPad Software).

**Correlative light and electron microscopy (CLEM)**. HeLa GFP-RUSC2 (clone 3) cells were mixed with wild-type HeLa cells and seeded on alpha-numeric gridded glass-bottom coverslips (P35G-1.5-14-C-GRID, MatTek) to be 40–50% confluent at the time of fixation. Cells were fixed with 2% formaldehyde/2.5% glutaraldehyde/ 0.1 M cacodylate buffer for 30 min at room temperature and washed with 0.1 M cacodylate. Cells were then stained with Hoechst (to stain the nucleus) for 2 min, before being washed with 0.1 M cacodylate. GFP-RUSC2 fluorescence signal was imaged on an LSM780 confocal microscope (63×/1.4 NA oil immersion objective; ZEISS) and the coordinates of cells selected for imaging were recorded. To prepare for electron microscopy cells were secondarily fixed with 1% osmium tetroxide/ 1.5% potassium ferrocyanide and then incubated with 1% tannic acid in 0.1 M cacodylate to enhance membrane contrast. Samples were washed with $dH_2O$ and dehydrated using increasing concentrations of ethanol. Epoxy resin (Araldite CY212 mix, Agar Scientific) was mixed at a 1:1 ratio with propylene oxide and this was used for 1 h to infiltrate the samples with resin, following which it was replaced with neat Epoxy resin. Pre-baked resin stubs were inverted over coordinates of interest, resin was cured overnight at 65 °C, following which stubs were removed from coverslips using liquid nitrogen. Areas of interest were identified by alpha-numeric coordinates and 70 nm ultrathin sections were collected using a Diatome diamond knife attached to an ultracut UCT ultramicrotome (Leica). As areas of interest were at the very basal surfaces of cells (and so the very top of the resin stub), sections were immediately collected onto piloform-coated slot grids. Sections were stained with lead citrate before being imaged on a Tecnai Spirit transmission electron microscope (FEI) at an operating voltage of 80 kV. HeLa GFP-RUSC2 and wild-type cells were imaged, and peripheral accumulations of uncoated vesicular and tubular structures were only observed in the regions of HeLa GFP-RUSC2 cells that correlated with the GFP fluorescence.

For the CLEM analysis of RUSC2 overexpressing cells in starvation conditions, HeLa RUSC2-GFP (clone 1) cells were seeded on alpha-numeric gridded glass-bottom coverslips, as above, and incubated in EBSS with 100 nM Bafilomycin A1 for 2 h before fixation. All other steps were performed as described above.

**Western blotting**. Estimations of protein concentrations were made using a Pierce BCA Protein Assay Kit (Thermo Fisher Scientific). Cells were lysed for Western blot analysis in 2.5% (w/v) SDS/50 mM Tris pH 8. Lysates were passed through a QIAshredder column (Qiagen) to shred DNA, incubated at 65 °C for 3 min and then boiled in NuPAGE LDS Sample Buffer (Thermo Fisher Scientific). Samples were loaded at equal protein amounts for SDS-PAGE, performed on NuPAGE 4–12% Bis–Tris gels in NuPAGE MOPS SDS Running Buffer, or for LC3B blots, on NuPAGE 12% Bis–Tris gels in NuPAGE MES SDS Running Buffer (all Thermo Fisher Scientific). PageRuler Plus Prestained Protein Ladder (Thermo Fisher Scientific) was used to estimate the molecular size of bands. Proteins were transferred to nitrocellulose membrane by wet transfer and membranes were blocked in 5% w/ v milk in PBS with 0.1% v/v Tween-20 (PBS-T). Primary antibodies (diluted in 5% milk) were added for at least 1 h at room temperature, followed by washing in PBS-T, incubation in secondary antibody (also in 5% milk) for 30 min at room temperature, washing in PBS-T and finally PBS. Chemiluminescence detection of HRP-conjugated secondary antibody was carried out using Amersham ECL Prime Western Blotting Detection Reagent (GE Healthcare) and X-ray film. Where

representative blots are shown, the experiment was repeated at least two times. Uncropped scans of all blots are shown in Supplementary Fig. 10–12.

**Protein expression and purification for GST pulldowns.** Human AP4E1 (residues 881–1135[18]) and AP4B1 (residues 612–739[26]) appendage domains were expressed as GST-fusion proteins in BL21(DE3)pLysS cells (Invitrogen) for 16–20 h at 22 °C after induction with 0.4 mM IPTG. Proteins were purified in 20 mM HEPES pH 7.5, 200 mM NaCl, and 2 mM 2-Mercaptoethanol. Cells were lysed using a disruptor (Constant Systems Limited) and proteins were affinity purified using glutathione sepharose (GE Healthcare). Fusion proteins were eluted in buffer with 30 mM reduced glutathione and further purified by gel filtration on a Superdex S200 preparative column (GE Healthcare).

**GST pulldowns from cytosol.** All steps were performed on ice with pre-chilled ice-cold buffers, unless otherwise noted.

To prepare cytosol for pulldowns, HeLa cells stably expressing EGFP or GFP-RUSC2 (clone 3), each in two 15 cm plates, were washed in PBS and then cytosol buffer (20 mM HEPES pH 7.5, 150 mM NaCl) and scraped in a total volume of 800 µL cytosol buffer plus 2 mM DL-Dithiothreitol (DTT, D5545, Sigma Aldrich). Cells were transferred to a 1 mL Dounce homogeniser (Wheaton) and homogenised with 15 strokes with the tight pestle, followed by 10 passes through a 30.5 gauge needle. Homogenates were then centrifuged at 78,400×g (RCF max) in a TLA-110 rotor (Beckman Coulter) for 30 min to pellet cell debris and the supernatants (cytosol) transferred to new tubes, snap frozen in liquid nitrogen, before storage at −80 °C until use in the pulldown assay.

For the pulldowns with AP-4 appendage domains, glutathione sepharose 4B (GE Healthcare) resin was batch equilibrated with 20 mM HEPES pH 7.5, 150 mM NaCl, 2 mM DTT. The 50% resin slurry (60 µL) was incubated and rotated with 50 µg of GST, GST-AP4E1, or GST-AP4B1 appendage for 1 h at 4 °C. Equal total protein amounts of HeLa lysates containing GFP-RUSC2 were added independently to each bait sample. Samples were incubated and rotated for 1 h at 4 °C. Samples were centrifuged at 2500×g for 5 min. Supernatant was removed, and resin was washed with 1 mL of wash buffer (20 mM HEPES pH 7.5, 150 mM NaCl, 2 mM DTT, 0.5% NP-40) for a total of three washes. After final removal of supernatant, 65 µL elution buffer (20 mM HEPES pH 7.5, 150 mM NaCl, 2 mM DTT, 60 mM reduced glutathione) were added to each sample and incubated for 10 min at 4 °C. Samples were centrifuged at 5000×g for 5 min, then supernatant was removed and transferred to a new tube. 10 µL fresh 1 M DTT and 25 µL SDS loading dye were added, and samples were boiled at 95 °C for 10 min. Control HeLa lysates containing GFP were treated in the same way as a control.

**Immunoprecipitations.** All steps were performed on ice with pre-chilled ice-cold buffers, unless otherwise noted.

For sensitive immunoprecipitation of TEPSIN-GFP, wild-type HeLa cells (control) and HeLa cells stably expressing TEPSIN-GFP[18] were grown in SILAC media. For two biological replicates (data shown in Fig. 2f) the wild-type cells were heavy labelled and the TEPSIN-GFP cells light labelled. A third replicate was performed with a label swap and used to filter out unlabelled proteins from the data. For each cell line, two 100 mm plates were washed once in PBS (without CaCl₂ and MgCl₂; 14190–094, Thermo Fisher Scientific) and then scraped in 4 mL PBS. Cells were transferred to a Dounce homogeniser (Sartorius) and homogenised with 20 strokes with the tight pestle, followed by two passes through a 21 gauge needle. Triton-TX-100 was added to the cells to a final concentration of 0.01%, cells were incubated at 4 °C for 20 min with rotation, and homogenates were cleared by centrifugation at 4000×g for 10 min. A portion of each homogenate was retained as input and the remainder was incubated with GFP-Trap A beads (ChromoTek) at 4 °C for 3 h with rotation. Beads were washed five times with 0.01% Triton-TX-100 and then immunoprecipitates were eluted in 100 µl 2.5% (w/v) SDS/50 mM Tris pH 8 and heated at 65 °C for 5 min. Beads were pelleted and supernatants (immunoprecipitates) transferred to new tubes. Equal volumes of control and TEPSIN-GFP immunoprecipitates were mixed and in-solution tryptic digest and peptide purification (single shot) were performed as described below.

For conventional immunoprecipitation[68], wild type or TEPSIN-GFP expressing cells[18] were SILAC labelled (in biological duplicate, with label-swap). For harvesting, cells from 2 × 500 cm² dishes (per IP) were washed twice in PBS, scraped in 10 mL of PBS-TT (PBS, 0.2% (v/v) Triton X-100, 0.1% (v/v) Tween-20), and incubated for 25 min with rotation. Insoluble material was removed by centrifugation at 4800×g for 3 min, followed by centrifugation at 21,000×g for 20 min. The supernatant was then further cleared by filtration through a 0.22 µm syringe filter. Lysates were pre-absorbed against Protein A sepharose beads. Approximately 60 µg of rabbit polyclonal anti-GFP antibody (a gift from Matthew Seaman) was added to cleared lysates. After incubation at 4 °C for 90 min with rotation, 50 µl of Protein A sepharose beads were added, and samples were incubated for a further 45 min. Beads were washed four times in PBS-T, once in PBS, and immunoprecipitates recovered in 100 µL Soft Elution Buffer (0.2% (w/v) SDS, 0.1% (v/v) Tween-20, 50 mM Tris-HCl pH 8) by incubation for 7 min at 25 °C. Eluates from TEPSIN-GFP expressing and wild type control cells were then combined prior to acetone precipitation and analysis by mass spectrometry.

For immunoprecipitation of GFP-RUSC2, wild-type and AP4B1 knockout HeLa cells stably expressing GFP-RUSC2 (mixed populations), and parental wild-type HeLa cells, in 100 mm plates were washed once in PBS and then scraped in 500 µL GFP-trap lysis buffer (10 mM Tris-HCl pH 7.5, 100 mM NaCl, 0.5 mM EDTA, 0.5% NP-40), supplemented with cOmplete™ EDTA-free protease inhibitor (Sigma-Aldrich). Cells were incubated on ice in lysis buffer for 10 min and then lysates were cleared by centrifugation at 16,000×g for 10 min. A protein assay was performed and, if required, lysates were adjusted to equal protein concentrations with lysis buffer. A portion of each lysate was retained as input and the remainder was incubated with GFP-Trap A beads at 4 °C for 3 h with rotation. Beads were washed five times with GFP-trap lysis buffer and then boiled in NuPAGE LDS Sample Buffer at 75 °C for 10 min to prepare for Western blot analysis.

**BioID streptavidin pulldowns.** All steps were performed on ice with pre-chilled ice-cold buffers, unless otherwise noted.

Streptavidin pulldowns for BioID were carried out from HeLa cells stably expressing BirA*-tagged AP4B1/E1/M1/S1 and control wild-type HeLa, HeLa BirA* and HeLa GFP-BirA* cells, in triplicate (experiments performed on three separate days). Cells were cultured in the presence of 50 µM biotin for 24 h prior to performing the experiment. Cells were harvested by scraping into 5 mL PBS, pelleted (600×g, 5 min) and washed twice in PBS. Lysis was performed in 1 mL RIPA buffer (TBS (50 mM Tris-HCl pH 7.4, 150 mM NaCl), 1% NP-40, 0.5% sodium deoxycholate, 1 mM EDTA, 0.1% SDS) supplemented with cOmplete™ EDTA-free protease inhibitor. DNA was sheared by running the lysates ten times through a 25 gauge needle, lysates were incubated at 4 °C for 10 min with mixing and then were sonicated (three times 5 s bursts with an amplitude of 10 µm). Lysates were cleared by centrifugation (16,000×g, 15 min) and supernatants transferred to new tubes and normalised to cell pellet weight with RIPA buffer. Biotinylated proteins were affinity purified using Pierce High Capacity Streptavidin Agarose (Thermo Fisher Scientific) by incubating with the lysates for 3 h with rotation at 4 °C. Beads were then pelleted and washed three times in RIPA buffer, twice in TBS and three times in 50 mM ammonium bicarbonate (ABC) pH 8 (09830, Sigma Aldrich), before resuspension in 50 mM ABC. Proteins were then reduced by the addition of 10 mM DTT at 56 °C for 30 min and alkylated by the addition of 55 mM iodoacetamide (I1149, Sigma Aldrich) at room temperature for 20 min in the dark. Proteins were enzymatically digested by addition of 1 µg Trypsin (V5280, Promega; stock at 0.1 mg mL⁻¹ in 1 mM HCl) and overnight incubation at 37 °C. The following day the beads were pelleted, tryptic peptides (supernatant) collected, spiked with 1 µL 100% trifluoroacetic acid (TFA), dried almost to completion in a centrifugal vacuum concentrator (Concentrator 5301, Eppendorf) and then stored at −20 °C. Later samples were thawed, resuspended in a total volume of 100 µL 1% (v/v) TFA and purified on SDB-RPS StageTips (single shot) as described below.

**Preparation of vesicle-enriched fractions.** For the comparative proteomic profiling of the vesicle fraction of AP-4-depleted cells, vesicle-enriched fractions were prepared from paired SILAC-labelled control and AP-4-depleted HeLa cell lines: wild-type versus AP4B1 knockout (two experiments), wild-type versus AP4E1 knockout (two experiments), wild type (untreated) versus AP-4 knockdown (three experiments), control (untreated) versus AP-4 knocksideways for 60 min (two experiments). Replicate experiments were performed on separate days.

All steps were performed on ice with pre-chilled ice-cold buffers, unless otherwise noted. For each sample eight confluent 15 cm dishes of cells were washed with PBS and scraped in a total volume of 7.5 mL Buffer A (0.1 M MES, pH 6.5, 0.2 mM EGTA and 0.5 mM MgCl₂). Cells were homogenised with 20 strokes of a motorised Potter-Elvehjem homogeniser (or a hand-held Dounce homogenizer with tight pestle) and cell debris was removed by centrifugation at 4150×g for 32 min. Supernatants were treated with 50 µg mL⁻¹ ribonuclease A (MP Biomedicals) for 1 h and then partially digested ribosomes were pelleted by centrifugation (4150×g for 3 min) and discarded. Membranes were pelleted by centrifugation at 55,000 rpm (209,900×g RCFmax) for 40 min in an MLA-80 rotor (Beckman Coulter). Membrane pellets were resuspended in 400 µL Buffer A using a 1 mL Dounce homogeniser, mixed with an equal volume of FS buffer (12.5% [w/v] Ficoll and 12.5% [w/v] sucrose, in buffer A) and centrifuged at 20,000 rpm (21,700×g RCFmax) for 34 min in a TLA-110 rotor, to pellet contaminants. Supernatants were diluted with four volumes of Buffer A and centrifuged at 40,000 rpm (86,700×g RCFmax) in a TLA-110 rotor for 30 min to obtain the vesicle-enriched fraction (pellet). Pellets were resuspended in 50 µL 2.5% SDS (in 50 mM Tris pH 8), heated at 65 °C for 3 min and centrifuged to pellet insoluble material (16,000×g, 1 min). For mass spectrometry, equal amounts of protein (20–50 µg) from paired SILAC-labelled control and AP-4-depleted vesicle fractions were mixed and either processed by in-solution or in-gel tryptic digest as described below.

**Generation of Dynamic Organellar Maps and membrane fractions.** Organellar maps were prepared from wild-type (control), AP4B1 knockout and AP4E1 knockout HeLa cells, in duplicate (six maps in total). Maps were prepared on two separate days, with a complete set of three on each occasion (one control, one AP4B1 knockout, and one AP4E1 knockout).

All steps were performed on ice with pre-chilled ice-cold buffers. HeLa cells (1 × 15 cm dish SILAC light and 1 × 15 cm dish SILAC heavy per map) were

washed in PBS (without $CaCl_2$ and $MgCl_2$), incubated in PBS for 5 min, washed in hypotonic lysis buffer (25 mM Tris-HCl pH7.5, 50 mM sucrose, 0.5 mM $MgCl_2$, 0.2 mM EGTA), and then incubated in hypotonic lysis buffer for 5 min. Cells were scraped in 4 mL hypotonic lysis buffer and mechanically lysed with 15 strokes of a Dounce homogeniser (8530700, tight pestle; Sartorius). The sucrose concentration was then restored to 250 mM. Lysates were centrifuged at 1000×g for 10 min to pellet nuclear material and post-nuclear supernatants were transferred to new tubes. The SILAC light post-nuclear supernatant was subfractionated into five fractions by a series of differential centrifugation steps: 3000×g for 10 min, 5400×g for 15 min, 12,200×g for 20 min, 24,000×g for 30 min, 78,400×g for 30 min (all speeds RCF max). All pellets were resuspended in 2.5% SDS/50 mM Tris pH 8 and heated for 5 min at 72˚C. In parallel, a single membrane fraction was obtained from the SILAC heavy post-nuclear supernatant by centrifugation at 78,400×g (RCF max) for 30 min. This fraction served as an internal reference, by spiking it into each of the "light" subfractions. Analysis by mass spectrometry provided a ratio of enrichment/depletion for each protein in each subfraction, relative to the standard. All five ratios combined yielded an abundance distribution profile for each protein across the subfractions. Principal component analysis revealed which proteins had similar fractionation profiles (apparent as organellar clusters in Fig. 1g–i). A detailed description of the method was previously published[19].

The membrane fractions analysed by label free quantification (data displayed in Fig. 2d) are the same as the heavy reference membrane fractions generated during the preparation of the Organellar maps, but with an additional biological replicate to give three biological replicates for each cell line.

**In-solution digestion of proteins.** Protein was precipitated by the addition of 5 volumes of ice-cold acetone, incubated at −20 °C for 30 min and pelleted by centrifugation at 4 °C for 5 min at 10,000×g. Precipitated protein was rinsed in ice-cold 80% acetone and re-pelleted as above. All subsequent steps were performed at room temperature. Precipitated protein pellets were air-dried for 5 min, resuspended in digestion buffer (50 mM Tris pH 8.1, 8 M Urea, 1 mM DTT) and incubated for 15 min. Protein was alkylated by addition of 5 mM iodoacetamide for 20 min and then enzymatically digested by addition of LysC (V1071, Promega; 1 mg per 50 mg of protein) for at least 3 h. Digests were then diluted four-fold with 50 mM Tris pH 8.1 before addition of Trypsin (1 mg per 50 mg of protein) for an overnight incubation. The peptide mixtures were then acidified to 1% (v/v) TFA in preparation for peptide purification and fractionation.

**Peptide purification and fractionation.** Several different peptide fractionation and clean-up strategies were used in this study. For most mass spectrometric experiments, peptides were purified and fractionated on SDB-RPS (#66886-U, Sigma) StageTips[69]. Peptide mixtures in 1% TFA were loaded onto activated StageTips and washed with Proteomics Wash Buffer (Preomics) and then 0.2% (v/ v) TFA. For single shot analysis, peptides were eluted with 60 μL Buffer X (80% (v/ v) acetonitrile, 5% (v/v) ammonium hydroxide). For triple-fractionation, peptides were eluted successively using 20 μL SDB-RPSx1 (100 mM ammonium formate, 40% (v/v) acetonitrile, 0.5% (v/v) formic acid), then 20 μL SDB-RPSx2 (150 mM ammonium formate, 60% (v/v) acetonitrile, 0.5% (v/v) formic acid), then 30 μL Buffer X. For six-fold fractionation, peptides were processed by strong cation exchange (SCX) on StageTips[69]. A detailed description of these methods has been published[69]. Alternatively, protein samples were separated by SDS-PAGE, gels were cut into 5–10 slices, and proteins were digested with trypsin in-gel. Peptide extracts were then cleaned up on $C_{18}$-StageTips, before elution in Buffer B (80% (v/v) acetonitrile, 0.5% (v/v) acetic acid)[70]. Cleaned peptides were dried almost to completion in a centrifugal vacuum concentrator, and then volumes were adjusted to 10 μL with Buffer A* (0.1% (v/v) TFA, 2% (v/v) acetonitrile) and either immediately analysed by mass spectrometry, or first stored at −20 °C.

The following techniques were applied for the various mass spectrometric analyses of this study: single shot SDB-RPS: BioID samples and sensitive IPs; triple-fractionation SDB-RPS: organellar maps analyses, membrane proteome analysis, vesicle prep analyses (AP-4 knockouts); six-fraction SCX: whole cell lysate full proteome analysis; in-gel digestion with multiple gel slice fractions: vesicle prep analyses (AP-4 knockdown, AP-4 knocksideways), conventional IPs.

**Mass spectrometry.** An overview of the mass spectrometric analyses performed during this study is provided in Supplementary Table 1. This includes information on quantification strategy, number of replicates, sample fractionation approach, MS instrument, and number of MS runs, for each analysis. Two different mass spectrometers were used (Q-Exactive HF[19,20] and Q-Exactive[71]; Thermo Fisher Scientific), as indicated in Supplementary Table 1.

**Processing of mass spectrometry data.** Mass spectrometry raw files were processed in MaxQuant[72] version 1.5, using the human SwissProt canonical and isoform protein database, retrieved from UniProt (www.uniprot.org). For SILAC experiments (vesicle fractions; Dynamic Organellar Map subfractions; whole cell lysate analysis; TEPSIN-GFP immunoprecipitations) multiplicity was set to 2, with Lys8 and Arg10 selected as heavy labels; Re-quantify was enabled; minimum number of quantification events was set to 1. For label-free experiments (membrane fractions; BioID) multiplicity was set to 1; LFQ was enabled, with LFQ

minimum ratio count set to 1. Membrane fractions were SILAC heavy labelled (Arg10, Lys8). Matching between runs was enabled. Default parameters were used for all other settings.

**Proteomic data analysis.** All analyses were performed on the 'protein groups' file output from MaxQuant. Data transformation, filtering and statistical analyses were performed in Perseus software[73] version 1.5 and Microsoft Excel. Principal component analysis (PCA) was performed in SIMCA 14 (Umetrics/MKS). PCA plots (Figs. 1g–i and 2b) show projections along 1st and 3rd principal components, for optimum visualisation. For all experiments identifications were first filtered by removing matches to the reverse database, matches based on modified peptides only, and common contaminants ('standard filtering'). Further experiment-specific filtering, data transformation and analyses were performed as described below.

**Dynamic Organellar Maps statistical analysis.** To identify proteins with shifted subcellular localisation in response to AP-4 knockout, we applied the rigorous statistical approach developed in our laboratory[19,20]. We adapted the procedure to the experimental design of the present study as follows. Organellar maps were made in duplicate, from control, *AP4B1* knockout and *AP4E1* knockout cells. Abundance distribution profiles across all six maps were determined for 3926 proteins. First, the profiles obtained in the *AP4B1* knockout and *AP4E1* knockout cells were subtracted from the profiles obtained in the cognate control map, protein by protein, to obtain 2 × 2 sets of delta profiles (Con_1-AP4B1_1, Con_2-AP4B1_2; Con_1-AP4E1_1, Con_2-AP4E1_2). For proteins that do not shift, the delta profile should be close to zero. All delta profile sets were subjected to a robust multivariate outlier test, implemented in Perseus software[73], to identify proteins with delta profiles significantly different from experimental scatter. The profile distance corresponds to a *p*-value reflecting how likely it is to observe this deviation by chance, assuming no real change. For each protein, four such *p*-values were hence obtained, two from *AP4E1* knockout, and two from *AP4B1* knockout. For maximum stringency, we selected the least significant of these *p*-values as representative of a protein's shift. A shift of equal or greater significance was thus observed in all four comparisons. We did not treat the four delta maps as completely independent though, since both knockouts were compared to the same cognate control. Hence, as a very conservative measure of movement, the selected *p*-value was only squared (instead of being raised to the power of four), and then corrected for multiple hypothesis testing using the Benjamini Hochberg Method. The negative $\log_{10}$ of the corrected *p*-value corresponded to a protein's movement (M) score.

Next, the reproducibility of observed delta profiles across replicates was determined as the Pearson correlation (Δmap (Con_1-AP4B1_1) vs. Δmap (Con_1-AP4B1_2); and Δmap (Con_1-AP4E1_1) vs. Δmap (Con_2-AP4E1_2)). For maximum stringency, we chose the lower one of the two obtained correlations as representative of the protein's shift reproducibility, corresponding to its R score.

To control the false discovery rate (FDR), we then applied the same analysis to our previously published wild-type HeLa maps[19] (six untreated maps with no genuine protein shifts expected). In this mock experiment, we designated two maps as controls, two as 'mock knockout 1', and two as 'mock knockout 2'. As above, we calculated M and R scores from the lowest correlations and p values of movement. The estimated FDR at a given set of M and R score cut offs was then calculated as the number of hits obtained with the mock experiment data, divided by the number of hits obtained with the AP-4 maps data, scaled by the relative sizes of the datasets (which were almost identical). At the chosen high stringency cut-offs (M-score > 4, R-score > 0.81), not a single hit was obtained from the mock data. Hence, we estimate the FDR for the three hits obtained from the AP-4 maps at <1%.

Finally, as an additional criterion, we also evaluated the similarity of identified shifts across the two different knockouts (i.e. the correlation of Δmap (Con_1-AP4B1_1) vs Δmap (Con_1-AP4E1_1); and Δmap (Con_2-AP4B1_2) vs Δmap (Con_2-AP4E1_2)). All three hits showed a very high degree of shift correlation (>0.9) across the two AP-4 knockout lines, thus also passing the additional stringency filter.

**Membrane fraction analysis.** Relative protein levels in membrane fractions from *AP4B1* knockout and *AP4E1* knockout HeLa cells (each in biological triplicate) were compared to those in membrane fractions from wild-type HeLa cells (in biological triplicate) using LFQ intensity data. The primary output was a list of identified proteins, and for each protein up to nine LFQ intensities across the wild-type and AP-4 knockout samples. Following standard data filtering, proteins were filtered to only leave those with nine LFQ intensities (no missing values allowed), leaving 6653 proteins. LFQ intensities were then log-transformed and comparison of knockout and wild-type membrane fractions performed with a two-tailed *t*-test. A permutation-based (1000 permutations) estimated FDR of 0.05 and an S0 parameter of 0.5 were set to define significance cut-offs (Perseus software).

**Whole cell lysate analysis.** Whole cell lysates from light-labelled *AP4B1* knockout and *AP4E1* knockout HeLa cells were compared to lysates from heavy-labelled wild-type HeLa cells by SILAC quantification, each in biological triplicate. The primary output was a list of identified proteins, and for each protein up to six H/L (Heavy/Light) ratios of relative abundance (three comparing *AP4B1* knockout to

wild-type and three comparing *AP4E1* knockout to wild-type). Following standard data filtering, proteins were further filtered to require at least two *H/L* ratios for each knockout, leaving 6841 proteins. *H/L* ratios from each replicate were then normalised to the median *H/L* ratio for that replicate, log-transformed, and inverted to *L/H* so that a protein depleted from the whole cell lysate in the absence of AP-4 had a negative ratio. A one-sample *t*-test (two-tailed) was applied to compare the *L/H* ratios for each protein to zero (null hypothesis of no change between wild-type and knockout). To control the false discovery rate (FDR), an identical analysis of a mock experiment comparing light-labelled and heavy-labelled wild-type HeLa lysates was performed (in triplicate; no genuine changes were expected here). The FDR was given by the number of hits observed in the mock experiment divided by the number of hits in the knockout experiment. Using the cut-offs $p \leq 0.02$ and a minimum absolute fold change ($\log_2$) of 0.45, the estimated FDR was 25%. The *t* scores from the mock experiment were calculated from only three datapoints per protein (whereas up to six were used for the knockout vs control data), and hence were adjusted to emulate a six datapoints experiment. To this end, we assumed that the observed standard deviations and means had been observed from six datapoints, yielding much lower *p* values from the same *t* scores for the mock data. This procedure thus likely overestimates the number of false positives at a given cut-off, resulting in highly stringent FDR control.

**Vesicle fraction analysis.** Paired AP-4-depleted and control vesicle fractions were compared by SILAC quantification. The primary output was a list of identified proteins, and for each protein up to nine *H/L* ratios of relative abundance, and the number of quantification events (*H/L* ratio count) used to calculate each ratio. Following standard data filtering, proteins were further filtered to require at least one *H/L* ratio count in all experiments. This excluded AP4S1 due to it having a ratio count of 0 in one experiment, so the data for AP4S1 was manually added back to the dataset (with one missing datapoint), giving a total of 2848 proteins quantified across all experiments. *H/L* ratios from each experiment were then normalised to the median *H/L* ratio for that experiment. For experiments in which the control cells were light labelled, normalised *H/L* ratios were inverted to *L/H*, and then all ratios were log transformed for plotting, so that depletion from the vesicle fraction in the absence of AP-4 was represented by a positive value. To identify proteins which were consistently lost from the vesicle fraction from AP-4 depleted cells, the normalised log-transformed SILAC ratios from all nine experiments were scaled to unit variance and combined by PCA.

**TEPSIN-GFP immunoprecipitations.** TEPSIN-GFP immunoprecipitations (conventional and sensitive) were compared to control immunoprecipitations by SILAC quantification. The primary output for each was a list of identified proteins, and for each protein one or more (up to three) *H/L* ratios of relative abundance between the TEPSIN-GFP and control immunoprecipitations, and the number of quantification events (*H/L* ratio count) used to calculate each ratio. Following standard data filtering, unlabelled proteins were filtered out based on having an *H/L* ratio < 0.2 in label-swapped replicates. Proteins were further filtered on a minimum *H/L* ratio count of 1 in all replicates, leaving 585 proteins for the conventional IP and 1128 proteins for the sensitive IP. *H/L* ratios from each replicate were normalised to the median *H/L* ratio for that replicate. For replicates in which the control cells were heavy labelled, normalised *H/L* ratios were inverted to *L/H*, and then all ratios were log transformed for plotting, so that enrichment in the TEPSIN-GFP immunoprecipitation was represented by positive values. The log SILAC ratios from two replicates were plotted against each other to reveal proteins enriched in the TEPSIN-GFP immunoprecipitations.

**AP-4 BioID.** Relative protein levels were compared across samples using LFQ intensity data. LFQ intensities from pulldowns from the control cell lines (HeLa, HeLa BirA* and HeLa GFP-BirA*) were compressed from nine values (three cell lines in triplicate) to three using control compression[74]. This creates a "worst-case scenario" control dataset where the three highest LFQ intensities are taken for each protein. Pulldowns from each AP-4 BioID cell line (in triplicate) were then compared to the compressed control dataset. Proteins were first filtered on LFQ intensities for valid values in all three replicate pulldowns from the AP-4 BioID cell line, leaving between approximately 3100–3700 proteins depending on the subunit. LFQ intensities were then log-transformed, and missing data points were imputed from a normal distribution with a downshift of 2.2 and a width of 0.3 standard deviations. Comparisons between control and AP-4 BioID cell lines were performed with a two-tailed *t*-test. A permutation-based (250 permutations) estimated FDR of 0.05 and an S0 parameter of 0.5 were set to define significance cut-offs (Perseus software).

## Data availability

The mass spectrometry proteomics data associated with Figs. 1 and 2 have been deposited to the ProteomeXchange Consortium [http://proteomecentral. proteomexchange.org] via the PRIDE partner repository[75] with the dataset identifier PXD010103. All other data supporting this work are available on reasonable request to the corresponding authors.

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

## Acknowledgements

We thank Matthias Mann for his continued support of this project, Grazia Mancini, Laurence Colleaux, Annick Raas-Rothschild, Xiao-Fei Kong, Jean-Laurent Casanova, and Stephanie Boisson-Dupuis for generously providing the patient cell lines, Marco Hein, Dick van den Boomen, Hayley Sharpe, and Tom O'Loughlin for advice, the CIMR microscopy core unit for technical expertise, Korbinian Mayr, Igor Paron, and Gabriele Sowa for outstanding technical support, Sebastian Schuck, Paul Luzio, Paul Manna, and Zuzana Kadlecova for critical reading of the manuscript, and CureSPG47 for inspiration. We give special thanks to all members of the Mann Department and Robinson Lab for valuable feedback. This work was funded by the German Research Foundation (DFG/Gottfried Wilhelm Leibniz Prize MA 1764/2–1), the Louis-Jeantet Foundation, the Max Planck Society for the Advancement of Science, a Wellcome Trust Principal Research Fellowship (086598) to M.S.R., an NIH grant (R35GM119525) to L.P.J., an NIHR Cambridge BRC PhD Fellowship to A.K.D., an EMBO Short-Term Fellowship to A.K.D., and a Wellcome Trust Strategic Award to the CIMR (100140). L.P.J. is a Pew Scholar in the Biomedical Sciences, supported by the Pew Charitable Trusts.

## Author contributions

G.H.H.B., M.S.R and A.K.D. conceptualised and designed the experiments. A.K.D. and G.H.H.B. performed and analysed most of the experiments. D.N.I. assisted in proteomic analyses and data visualisation. J.R.E. performed and analysed the CLEM. L.P.J. and T.L. A. performed protein purifications and pulldowns. J.H. assisted in the generation of the knocksideways cell line and provided the patient fibroblasts. A.K.D., G.H.H.B. and M.S.R

wrote the original and revised manuscripts, with contributions from the other authors. G.H.H.B. and M.S.R. supervised the project.

## Additional information

**Competing interests:** The authors declare no competing interests.

