## [Peer Review File · Nature Communications]

Reviewers' comments:

Reviewer #1 (Remarks to the Author):

The function of AP-4 is largely unknown although several cargo proteins were already identified and the deficiency of AP-4 was already known to induce the dysregulated autophagy in neurons. It was also known the deficiency of AP-4 induces the severe neurological disorder, but the pathological mechanism is unknown. In this manuscript, Davies et al identified three cargo protein and two accessory protein of AP-4 by using unbiased proteomic methods. They also indicated that the AP-4 accessory protein RUSC2 facilitated the microtubule plus-end directed transport of AP-4 derived vesicle. Basically, this is an important and interesting paper to understand both of the function of AP-4 pathway and the pathological mechanism of AP-4 deficiency. However, there are several points that should be answered before publishing in Nature Communications.

1. In Figure 1, the distribution of ATG9A shift from endosome to TGN by knocking out of AP-4. Figure 4 and 8 also suggested that the protein amount of ATG9A in TGN was increased in AP-4 deficient cells. However, the protein amount in other organelle may also be increased. Especially in Figure 8, the cells from the patients have higher expression level of ATG9A protein. Therefore Relative ATG9A overlap with endosome and ER markers should be examined.

2. In Figure 5, the authors indicated that the over expression of RUSC2 induced the relocation of ATG9A, SERINC1 and SERINC3. The relationship between the relocation of cargo proteins and autophagy should be analyzed. It is necessary to examine whether the over expression of RUSC2 induce the dysregulated autophagy or not by using immunoblot and immunocytochemical analysis as in Figure 7.

It is also necessary to clarify whether the interaction between RUSC and AP-4 play function in the transport of ATG9a and in autophagy. Overexpression of appendage domain of AP-4 should block the interaction between RUSC and AP-4. In this condition, the distribution of ATG9A and the amount of LC3B-II should be examined.

3. In their model shown in Figure 8c, they argued that the AP-4 and its accessory proteins concentrate ATG9a, SERINC1 and SERINC2 into a vesicle bud at the TGN. However, in the AP-4 deficient cells, SERINC proteins still localized in endosomes not in TGN, although ATG9A accumulated in TGN. How they can explain this discrepancy? One possibility is that the SERINC proteins can be the cargo of other Adaptor Proteins, such as AP-1. Therefore other adaptor proteins should be knocked down as well as AP-4 and the localization of SERINC proteins should be analyzed. Moreover, it is also necessary to examine what kind of endosome (i.e. early endosome, late endosome, recycling endosome) contains SERINC proteins in wild type and AP-4 deficient cells.

Reviewer #2 (Remarks to the Author):

Nature communications manuscript submission NCOMMS-17-33553-T by Davies and colleagues. For a long time, the mechanism of function and the scope of cargoes that are transported via AP-4 vesicles had remained unresolved. The current study makes an important contribution in this context. Using a number of highly sophisticated unbiased proteomics approaches, the authors not only identify AP-4 cargo proteins, ATG9A, SERINC1 and 3, but also two trafficking factors, RUSC1 and 2, which likely link AP-4 vesicles to microtubule transport machinery. Importantly, the sensitive proteomics techniques that were used here allowed the authors to convincingly conclude on an interaction of AP-4 with these cargoes and accessory proteins, where a previous study by Mattera et al., PNAS 2017 had failed to do so. The current study therefore largely goes beyond this earlier work, while at the same time laying to rest what one might have called the AP-4 conundrum. The experimental work is extremely thorough, and the manuscript is very well written and pleasant to read. The link between model cell systems and disease relevant cells is a highlight of this study. The authors should be commended for this brilliant contribution.

Reviewer #3 (Remarks to the Author):

This manuscript entitled "AP-4 vesicles unmasked by organellar proteomics to reveal their cargo and machinery" by Davies et. al. conducted Dynamic Organellar Maps to identify the potential cargo of AP-4 vesicles. As a results ATG9A, SERINC1 and SERINC3 showed significant localization

shift depending on AP-4. They also found two AP-4 accessory proteins Rusc1 and 2 by comparative vesicle profiling. AP-4 deficiency caused mislocalization of ATG9A in TGN, and leading to abnormal autophagosome formation. They also showed that overexpression of RUSC2 leads to mislocalization of ATG9A positive vesicles to the cell periphery, which is microtubule dependent.

Overall, the data are clean and convincing and this study will be important contribution to the field. However unfortunately, main finding that AP-4 dependent ATG9A traffic has been scooped by Bonifacino group recently, which reduces the enthusiasm of this study. To rationalize the publication in this journal, this referee requests the following points.

Major points;

1. Association of RUSC1/2 to AP-4 is the original of this study, and the associated results should be strengthened. What will happen when RUSC1/2 are knock down/out, especially regarding ATG9A localization and autophagy.
2. Figure 7C. What does the large LC3 signal represent? Does each autophagosome become larger? Or normal size autophagosome are accumulated? CLEM analysis is needed to clarify this point. According the obtained result, possible underlying mechanism needed to be discussed.
3. Figure 6. Does the overexpression of RUSC1 affect autophagy? Both flux assay and LC3 staining are needed.

Reviewer #1:

The function of AP-4 is largely unknown although several cargo proteins were already identified and the deficiency of AP-4 was already known to induce the dysregulated autophagy in neurons. It was also known the deficiency of AP-4 induces the severe neurological disorder, but the pathological mechanism is unknown. In this manuscript, Davies et al identified three cargo protein and two accessory protein of AP-4 by using unbiased proteomic methods. They also indicated that the AP-4 accessory protein RUSC2 facilitated the microtubule plus-end directed transport of AP-4 derived vesicle. Basically, this is an important and interesting paper to understand both of the function of AP-4 pathway and the pathological mechanism of AP-4 deficiency.

However, there are several points that should be answered before publishing in Nature Communications.

1. In Figure 1, the distribution of ATG9A shift from endosome to TGN by knocking out of AP-4. Figure 4 and 8 also suggested that the protein amount of ATG9A in TGN was increased in AP-4 deficient cells. However, the protein amount in other organelle may also be increased. Especially in Figure 8, the cells from the patients have higher expression level of ATG9A protein. Therefore Relative ATG9A overlap with endosome and ER markers should be examined.

ATG9A has a complex and well-documented subcellular distribution, with transient residence in all major organelles of the secretory pathway. We agree with the reviewer that the levels of ATG9A are increased in the patient fibroblast cell lines, as discussed in the manuscript. While the higher expression will most likely lead to an increase in ATG9A levels in multiple organelles, the most interesting observation from the point of this study is the accumulation of ATG9A at the TGN. Notwithstanding the obvious increase in overall staining, this relative increase in the TGN is very pronounced in all cells types we investigated. Given that AP-4 functions in TGN export, this is the feature most relevant to the pathway. Interestingly, there was no significant increase in ATG9A expression in AP-4 knockout HeLa cells (Supplemental Figure 1c), but we still observed the primary phenotype of ATG9A accumulation at the TGN (Figure 3a, b) and dysregulation of autophagy (Figure 8a-d). Increased levels of ATG9A in other organelles are thus separable from the primary defects, and not the main focus of this study; we hence decided not to pursue this further. However, we have modified the relevant passage in the text to make it clear that the overall increase in expression clearly results in higher levels of ATG9A in several organelles, and not just the TGN.

2. In Figure 5, the authors indicated that the over expression of RUSC2 induced the relocation of ATG9A, SERINC1 and SERINC3. The relationship between the relocation of cargo proteins and autophagy should be analyzed. It is necessary to examine whether the over expression of RUSC2 induce the dysregulated autophagy or not by using immunoblot and immunocytochemical analysis as in Figure 7.

It is also necessary to clarify whether the interaction between RUSC and AP-4 play function in the transport of ATG9a and in autophagy. Overexpression of appendage domain of AP-4 should block the interaction between RUSC and AP-4. In this condition, the distribution of ATG9A and the amount of LC3B-II should be examined.

We have followed the reviewer's recommendations and investigated the effects of both RUSC loss of function and RUSC overexpression on autophagy.

First, we overexpressed GFP-tagged RUSC2, in non-starved or starved HeLa cells, and analysed the effects on LC3B by Western blot (Supplemental Figure 8a, b). Overexpression of

RUSC2 had no obvious effect on LC3B-I or LC3B-II levels. Consistent results were obtained for RUSC2 tagged at either the C- or N-terminus.

Next, we investigated the effect of RUSC2 overexpression by immunofluorescence microscopy (Supplemental Figure 8c). As before, we observed accumulation of RUSC2 in peripheral clusters (corresponding to ATG9A vesicles). There were no obvious changes in the LC3B pattern in non-starved cells. However, under starvation conditions (which induce autophagy) RUSC2 puncta were distinctly less peripheral, although they were still positive for ATG9A. These RUSC2/ATG9A-positive puncta were often in very close proximity to LC3B-positive puncta (which appeared larger than in wild-type cells; Figure 9a). Enhanced resolution Airyscan microscopy revealed that the RUSC2 puncta were indeed distinct from the closely juxtaposed LC3B-positive structures (Figure 9b). To further investigate their ultrastructure, we imaged RUSC2-GFP puncta in starved cells with correlative light and electron microscopy (CLEM). This revealed frequent clusters of small vesicles surrounding morphologically identifiable autophagosomes (Figure 9c). Hence, our data suggest a model in which ATG9A- and RUSC2-positive vesicles appear to have a role in 'feeding' or perhaps seeding autophagosomes, thus providing a potential mechanism for spatial regulation of autophagy. We have incorporated these new mechanistic insights into our model of AP-4 function.

We then investigated the RUSC loss-of-function phenotype. The referee suggested overexpression of the AP-4 appendage domain to act as a dominant-negative. While conceptually sound, we decided against this approach, for several reasons. First, the unstructured regions of the AP-appendage domains tend to be rather unstable; second, the RUSC-AP4 interaction may be mediated by more than one binding site/subunit; third, we do not know how RUSC1 interacts with AP-4, and RUSC1 (as we found out) is functionally at least partially redundant with RUSC2; and fourth, the appendage domains have at least one other binding partner, tepsin, so dominant negative effects could reflect loss of function of tepsin rather than RUSC2. Therefore, we used a more direct depletion strategy by siRNA mediated knockdown. Since there are no commercial antibodies available for detection of either RUSC1 or RUSC2, we had to rely on quantitative RT-PCR to gauge knockdown efficiency. The RUSC1 knockdown was relatively straightforward. For RUSC2, we tested eight different siRNAs (from two different vendors); only two of them had reasonable knockdown efficiency (Figure 8f and data not shown). We then assessed the effects of RUSC depletion on autophagy by Western analysis of LC3B, as suggested by the reviewer. While RUSC1 depletion alone had no obvious phenotype, RUSC2 depletion strongly changed the pattern of LC3B staining, with substantially higher overall levels of LC3B, and a shifted ratio of LC3B-I to LC3B-II (Figure 8g). This phenotype is strongly reminiscent of the autophagy dysregulation observed in AP-4 knockout cells (Figure 8a, b). Furthermore, a combined knockdown of RUSC1 and RUSC2 had an even stronger effect than RUSC2 depletion alone, demonstrating that RUSC1 is also functionally related to RUSC2. Our new data clearly demonstrate that RUSCs are important components of the AP-4 pathway. Importantly, the RUSC2 knockdown phenotype was highly consistent with both independent RUSC2 siRNAs oligos (as stated in the methods); Figure 8g shows the result for only one of the oligos though, to keep the figure simple.

[redacted]

3. In their model shown in Figure 8c, they argued that the AP-4 and its accessory proteins concentrate ATG9A, SERINC1 and SERINC2 into a vesicle bud at the TGN. However, in the AP-4 deficient cells, SERINC proteins still localized in endosomes not in TGN, although ATG9A accumulated in TGN. How they can explain this discrepancy? One possibility is that the SERINC proteins can be the cargo of other Adaptor Proteins, such as AP-1. Therefore other adaptor proteins should be knocked down as well as AP-4 and the localization of SERINC proteins should be analyzed. Moreover, it is also necessary to examine what kind of endosome (i.e. early endosome, late endosome, recycling endosome) contains SERINC proteins in wild type and AP-4 deficient cells.

Even in wild-type cells SERINC proteins and ATG9A have only partially overlapping steady state localisations, and most likely largely different intracellular itineraries. Although all three proteins clearly co-localise in TGN-derived, AP-4-dependent vesicles, they also localise to multiple other compartments, including the plasma membrane and endosomes. This strongly suggests that they can be transported by multiple types of transport intermediates. For ATG9A there is a substantial body of literature supporting this view, and the cytosolic domains of SERINC proteins harbour consensus motifs for interactions with clathrin adaptors. In the absence of AP-4, both SERINC proteins and ATG9A can clearly still exit the TGN, but with different efficiencies – for ATG9A the AP-4 pathway seems to be more important. In our opinion, the different distributions observed for ATG9A and SERINC proteins in AP-4 knockout cells are therefore not surprising. Nevertheless, we agree with the reviewer that this is an important and potentially confusing point, and we have added a clarifying sentence to the results section.

Regarding the reviewer's suggestion to characterise the other export routes functionally, we feel that this would distract from the focus of the present study, which is the AP-4 pathway, and not the sorting of SERINC proteins and ATG9A in general. We have since not pursued this further.

Reviewer #2:

Nature communications manuscript submission NCOMMS-17-33553-T by Davies and colleagues. For a long time, the mechanism of function and the scope of cargoes that are transported via AP-4 vesicles had remained unresolved. The current study makes an important contribution in this context. Using a number of highly sophisticated unbiased

proteomics approaches, the authors not only identify AP-4 cargo proteins, ATG9A, SERINC1 and 3, but also two trafficking factors, RUSC1 and 2, which likely link AP-4 vesicles to microtubule transport machinery. Importantly, the sensitive proteomics techniques that were used here allowed the authors to convincingly conclude on an interaction of AP-4 with these cargoes and accessory proteins, where a previous study by Mattera et al., PNAS 2017 had failed to do so. The current study therefore largely goes beyond this earlier work, while at the same time laying to rest what one might have called the AP-4 conundrum. The experimental work is extremely thorough, and the manuscript is very well written and pleasant to read. The link between model cell systems and disease relevant cells is a highlight of this study. The authors should be commended for this brilliant contribution.

This reviewer did not request any alterations to the manuscript.

Reviewer #3:

This manuscript entitled "AP-4 vesicles unmasked by organellar proteomics to reveal their cargo and machinery" by Davies et. al. conducted Dynamic Organellar Maps to identify the potential cargo of AP-4 vesicles. As a results ATG9A, SERINC1 and SERINC3 showed significant localization shift depending on AP-4. They also found tow AP-4 accessory proteins Rusc1 and 2 by comparative vesicle profiling. AP-4 deficiency caused mislocalization of ATG9A in TGN, and leading to abnormal autophagosome formation. They also showed that overexpression of RUSC2 leads to mislocalization of ATG9A positive vesicles to the cell periphery, which is microtubule dependent.

Overall, the data are clean and convincing and this study will be important contribution to the field. However unfortunately, main finding that AP-4 dependent ATG9A traffic has been scooped by Bonifacino group recently, which reduces the enthusiasm of this study. To rationalize the publication in this journal, this referee requests the following points.

Major points;

1. Association of RUSC1/2 to AP-4 is the original of this study, and the associated results should be strengthened. What will happen when RUSC1/2 are knock down/out, especially regarding ATG9A localization and autophagy.

We have done the experiments as suggested by the reviewer – see above, Reviewer 1, point 2.

2. Figure 7C. What does the large LC3 signal represent? Does each autophagosome become larger? Or normal size autophagosome are accumulated? CLEM analysis is needed to clarify this point. According the obtained result, possible underlying mechanism needed to be discussed.

We agree with the reviewer that the reason underlying the apparent increase in autophagosomal size in the AP-4 knockout is an important point. According to the published literature LC3B levels correlate with autophagosome size, and since we see an increase in overall LC3B levels, the most likely scenario is that the autophagosomes are indeed larger, possibly due to a maturation defect. This is also the conclusion reached in the competing AP-4 publication cited by the reviewer (Mattera at el., 2017). Unfortunately, the suggested CLEM experiments are not possible as part of this revision. CLEM is technically very challenging and time consuming. We were able to perform the CLEM for RUSC2

overexpressing cells under starvation conditions within the three months revision time, since we already had cell lines stably expressing the protein of interest with a GFP tag; but even that was a major endeavour. To investigate autophagosome ultrastructure we would have to generate and functionally validate LC3B-GFP cell lines, with endogenous expression levels, in both wild-type and AP-4 knockout backgrounds, even before attempting the CLEM itself; this would be beyond the scope of this study. Nevertheless, we have taken up the reviewer's point and modified the discussion to include a comment about the increase in size.

3. Figure 6. Dose the overexpression of RUSC1 affect autophagy? Both flux assay and LC3 staining are needed.

The reviewer probably refers to RUSC2 overexpression (since RUSC1 overexpression has no effect on ATG9A localisation, unlike RUSC2 overexpression).

We have done the experiment as suggested – see above, Reviewer 1, point 2.

REVIEWERS' COMMENTS:

Reviewer #1 (Remarks to the Author):

The authors have answered all of my concerns by carrying out several experiments or by changing the manuscript. Now, I feel this paper is suitable to publish in Nature Communications.

Reviewer #3 (Remarks to the Author):

The authors responded appropriately to the comments I raised. Now the paper is acceptable.